# Endosome and Golgi-associated degradation (EGAD) of membrane proteins regulates sphingolipid metabolism

Oliver Schmidt[1] (iD), Yannick Weyer[1], Verena Baumann[1,†], Michael A Widerin[1], Sebastian Eising[2], Mihaela Angelova[3] (iD), Alexander Schleiffer[4,5] (iD), Leopold Kremser[6], Herbert Lindner[6], Matthias Peter[7], Florian Fröhlich[2] (iD) & David Teis[1,*] (iD)

## Abstract

Cellular homeostasis requires the ubiquitin-dependent degradation of membrane proteins. This was assumed to be mediated exclusively either by endoplasmic reticulum-associated degradation (ERAD) or by endosomal sorting complexes required for transport (ESCRT)-dependent lysosomal degradation. We identified in *Saccharomyces cerevisiae* an additional pathway that selectively extracts membrane proteins at Golgi and endosomes for degradation by cytosolic proteasomes. One endogenous substrate of this endosome and Golgi-associated degradation pathway (EGAD) is the ER-resident membrane protein Orm2, a negative regulator of sphingolipid biosynthesis. Orm2 degradation is initiated by phosphorylation, which triggers its ER export. Once on Golgi and endosomes, Orm2 is poly-ubiquitinated by the membrane-embedded "Defective in SREBP cleavage" (Dsc) ubiquitin ligase complex. Cdc48/VCP then extracts ubiquitinated Orm2 from membranes, which is tightly coupled to the proteasomal degradation of Orm2. Thereby, EGAD prevents the accumulation of Orm2 at the ER and in post-ER compartments and promotes the controlled de-repression of sphingolipid biosynthesis. Thus, the selective degradation of membrane proteins by EGAD contributes to proteostasis and lipid homeostasis in eukaryotic cells.

**Keywords** endosomes; Golgi; proteasome; sphingolipids; ubiquitin
**Subject Categories** Membrane & Intracellular Transport; Post-translational Modifications, Proteolysis & Proteomics
**The EMBO Journal (2019) 38: e101433**

See also: **D Fonseca & P Carvalho** (August 2019)

## Introduction

The function of cells depends on the integrity of their proteome. Proteomes of eukaryotic cells are estimated to consist of tens of millions of proteins (e.g., $50 \times 10^6$ proteins in *S. cerevisiae*) (Ghaemmaghami *et al*, 2003; Milo, 2013). Among those, cellular quality control networks detect and selectively degrade proteins that mis-fold, cannot integrate into protein complexes, fail to target to the correct organelles, or are destined for degradation by regulatory mechanisms (Juszkiewicz & Hegde, 2018). Thereby, selective protein degradation maintains proteome homeostasis (proteostasis). Chronic defects in proteostasis result in the toxic accumulation of proteins and cause cell injuries that are central to the pathophysiologies of human diseases, including cancer, autoimmunity, diabetes, obesity, and neurodegeneration. How selective protein degradation pathways function is therefore a major biological question with relevance to pathologies (Klaips *et al*, 2018).

In eukaryotic cells, integral membrane proteins are known to be selectively degraded by two mechanistically distinct pathways. At the endoplasmic reticulum (ER), membrane proteins are targeted for degradation by the ER-associated degradation (ERAD) pathways. ERAD is executed by membrane-embedded multi-subunit E3 ubiquitin ligase complexes. Their function is well characterized in *S. cerevisiae*. The ubiquitin ligases Hrd1 and Doa10 assemble into two different ERAD complexes, each with a different but also partially overlapping substrate spectrum (Hampton *et al*, 1996; Bordallo *et al*, 1998). The Doa10 complex detects defects in protein domains that are exposed to the cytosol (ERAD-C). The Hrd1 complex detects defects in luminal domains (ERAD-L) or in the transmembrane domains of proteins (ERAD-M) (Carvalho *et al*, 2006). Once substrates are detected, Hrd1 or Doa10 complexes mediate their

1 Division of Cell Biology, Biocenter, Medical University of Innsbruck, Innsbruck, Austria
2 Department of Biology/Chemistry, University of Osnabrück, Osnabrück, Germany
3 INSERM, Laboratory of Integrative Cancer Immunology, Sorbonne Université, Sorbonne Paris Cité, Université Paris Descartes, Centre de Recherche des Cordeliers, Université Paris Diderot, Paris, France
4 Research Institute of Molecular Pathology (IMP), Vienna Biocenter (VBC), Vienna, Austria
5 Institute of Molecular Biotechnology of the Austrian Academy of Sciences (IMBA), Vienna Biocenter (VBC), Vienna, Austria
6 Division of Clinical Biochemistry, Protein Micro-Analysis Facility, Biocenter, Medical University of Innsbruck, Innsbruck, Austria
7 Institute of Biochemistry, ETH-Zürich, Zurich, Switzerland
*Corresponding author. Tel: +43 512 9003 70191; Fax: +43 512 9003 73100; E-mail: david.teis@i-med.ac.at
†Present address: MFPL, University of Vienna, Vienna, Austria

retro-translocation across the ER membrane and ubiquitination. The ubiquitinated substrates are then extracted into the cytoplasm by the AAA-ATPase VCP/Cdc48 in an ATP-dependent manner and transferred in a soluble state to proteasomes for degradation (Hiller *et al*, 1996; Jarosch *et al*, 2002; Neuber *et al*, 2005; Schuberth & Buchberger, 2005; Wu & Rapoport, 2018). Yeast Doa10 is related to March6/TEB4 (Swanson *et al*, 2001), and yeast Hrd1 is related to human Hrd1 and gp78. Several other membrane-bound E3 ligases have been implicated in mammalian ERAD (Rnf5, Trc8, Rfp2, Rnf170, Rnf185), but compared to Hrd1, their function is still poorly characterized (Ruggiano *et al*, 2014).

ERAD can also degrade proteins in a regulated manner. A prominent regulated ERAD substrate is the 3-hydroxy-3-methylglutaryl coenzyme A reductase (HMGR), a key enzyme in sterol biosynthesis (Gil *et al*, 1985; Hampton *et al*, 1996). HMGR degradation is part of a feedback inhibition that is critical for sterol homeostasis in yeast and humans. An additional branch of ERAD operates at the inner nuclear membrane (INM) and is mediated by the Asi complex (Foresti *et al*, 2014; Khmelinskii *et al*, 2014).

Once membrane proteins are exported from the ER, they are ubiquitinated by different quality control systems (Arvan *et al*, 2002; Wang & Ng, 2010; Zhao *et al*, 2013). It is thought that all ubiquitinated membrane proteins in post-ER compartments (Golgi, endosomes, plasma membrane, lysosomes/vacuoles) are sorted by the endosomal sorting complexes required for transport (ESCRT) into the lumen of lysosomes for degradation. ESCRT-dependent membrane protein degradation is a multi-step process that involves the recognition of ubiquitinated membrane proteins and a reverse membrane budding reaction, and occurs typically via the so-called multivesicular body (MVB) pathway (Schmidt & Teis, 2012). In some cases, the ESCRT machinery can also bud cargo-laden vesicles from the limiting membrane of the vacuole directly into the lumen of the organelle (Zhu *et al*, 2017). The evolutionary conserved hallmark of ESCRT-deficient eukaryotic cells is the accumulation of ubiquitinated membrane proteins on endosomes, which is also associated with age-related neurodegenerative diseases (Skibinski *et al*, 2005; Filimonenko *et al*, 2007; Lee *et al*, 2007; Ren *et al*, 2008; Urwin *et al*, 2010).

The key role of the ESCRT pathway in cellular membrane protein degradation led us to explore whether eukaryotic cells employ additional mechanisms to reduce the accumulation of membrane proteins in post-ER compartments. Using a combination of genome-wide screens and quantitative proteomic profiling, we identified the endosome and Golgi-associated degradation (EGAD) pathway. Central to the function of EGAD is the membrane-embedded "Defective in SREBP cleavage" (Dsc) ubiquitin ligase complex. It selectively ubiquitinates integral membrane proteins at Golgi and endosomes. Unexpectedly, some of these ubiquitinated membrane proteins are not sorted by the ESCRT machinery into lysosomes. Instead, they are extracted from post-ER membranes by Cdc48/VCP in a step that is tightly linked to their degradation at cytosolic proteasomes. Candidate substrates of the EGAD pathway include ER-resident membrane proteins that are required for lipid biosynthesis. We have characterized the EGAD pathway in detail using Orm2, a negative regulator of sphingolipid (SL) biosynthesis, as a model substrate. Thereby, we establish the molecular steps required for EGAD-mediated membrane protein degradation. Furthermore, we demonstrate that the EGAD-dependent degradation of Orm2 is

essential for the homeostatic regulation of SL metabolism. These findings provide the basis for detailed mechanistic studies on the EGAD pathway and show that the ERAD, EGAD, and ESCRT pathways have distinct functions that collectively maintain eukaryotic membrane protein homeostasis.

# Results

## ESCRT-independent functions of the Dsc complex in membrane protein degradation

We hypothesized that genes essential for the survival of ESCRT mutants encode hitherto unknown mechanisms that reduce the intracellular accumulation of membrane proteins as part of the cellular proteostasis network. To identify these additional mechanisms, we disrupted in budding yeast the AAA-ATPase Vps4 of the ESCRT machinery (Babst *et al*, 1997) and used synthetic genetic interaction array (SGA) screening. The screen identified 119 non-essential genes that were required for the growth and survival of *vps4Δ* mutants (Fig 1A and Appendix Table S1). Gene Ontology term enrichment analysis was used to group these genes into known macromolecular protein complexes (Fig 1B and Appendix Table S2).

The autophagic machinery, ubiquitin ligase complexes of the ERAD system, or the associated unfolded protein response (UPR) did not show synthetic genetic interactions. Instead, another membrane-embedded E3 ubiquitin ligase complex, the defective in SREBP cleavage (Dsc) complex (Stewart *et al*, 2011; Tong *et al*, 2014), was among the top enriched protein complexes that contributed to the survival of ESCRT mutants (Fig 1A and B). The Dsc complex consists of four membrane-embedded subunits, Tul1, Ubx3, Dsc2, and Dsc3. Our screen identified *TUL1* and *UBX3,* and deleting each subunit of the Dsc complex caused synthetic growth defects with *vps4Δ* in a different genetic background (SEY6210 strain) at 26 and 37°C (Figs 1C and EV1A). Re-expression of *TUL1, UBX3,* or *VPS4* restored the growth defects of the respective double mutants (Fig 1C).

The functional domains of the Dsc subunits are highly conserved in vertebrates, and they resemble subunits of ERAD ubiquitin ligase complexes of the Hrd1/gp78 family (Fig 1D; Lloyd *et al*, 2013). Dsc2 is a rhomboid-like pseudo-protease orthologous to human ubiquitin-associated domain-containing protein 2 (UBAC2); the domain architecture of Dsc3 is similar to the transmembrane and ubiquitin-like domain-containing protein 1/2 (TMUB1/2) protein family (see Materials and Methods for details); Ubx3 harbors a Cdc48-recruiting UBX domain, and its thioredoxin-like domain is related to FAS-associated factor 2 (FAF2) (Christianson *et al*, 2011; Greenblatt *et al*, 2011; Lloyd *et al*, 2013; Olzmann *et al*, 2013; Voinciuc *et al*, 2013; Tong *et al*, 2014). Tul1 is a multi-spanning membrane protein with seven predicted transmembrane domains and a C-terminal cytosolic RING E3 ligase domain. Tul1 is conserved in a wide range of fungi and plants (e.g., Fly1 in *Arabidopsis thaliana*) (Reggiori & Pelham, 2002; Stewart *et al*, 2011; Voinciuc *et al*, 2013), in Alveolata (*Plasmodium*), Mycetozoa (*Dictyostelium*), and Stramenopiles (such as the diatom *Thalassiosira*) (Fig EV1B–D). A clear orthologue of Tul1 is missing in humans, but homology searches using a sequence model of the Tul1 RING domain

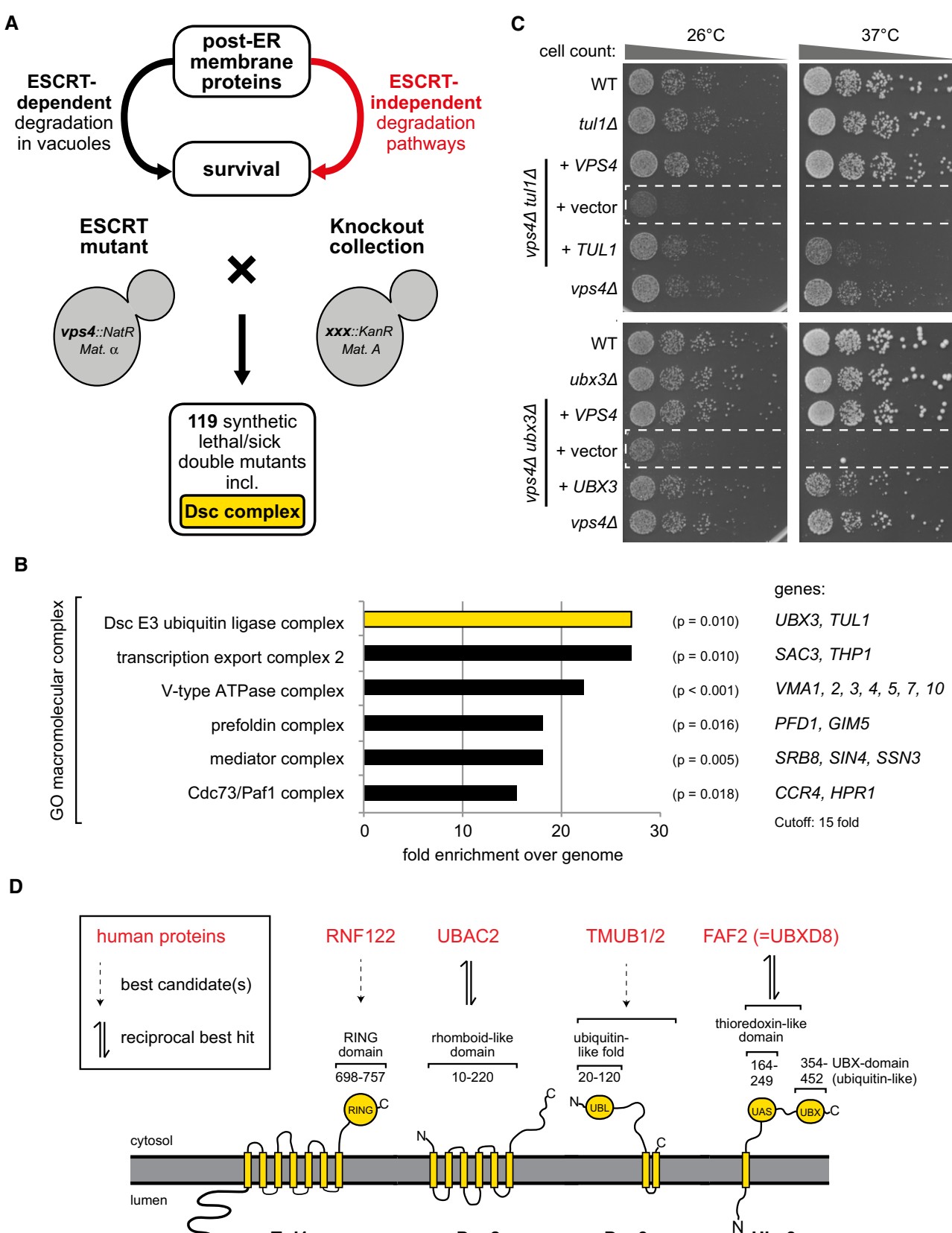

**Figure 1.**

◀

**Figure 1. ESCRT-independent functions of the Dsc complex in membrane proteostasis.**

See also Fig EV1 and Appendix Tables S1–S3.

A   Synthetic genetic array technology of *Saccharomyces cerevisiae* was used to cross the ESCRT mutant (query mutation *ups4::NatR*) with all non-essential genes (*xxx:: KanR*). The growth of double mutants (*ups4::NatR xxx::KanR*) was scored in two independent replicates for normal growth, synthetic sick, or synthetic lethal phenotypes. 119 genes (Appendix Table S1) showed synthetic sickness in one replicate and synthetic lethality in the other replicate or synthetic lethality in both replicates.

B   Gene Ontology (GO) analysis for macromolecular complexes of the 119 genes that showed synthetic sickness in one replicate and synthetic lethality in the other replicate or synthetic lethality in both replicates (Appendix Tables S1 and S2). A hypergeometric test was used to estimate whether the mapped GO term is significantly enriched with the selected genes. Significant GO terms ($P < 0.05$) and their associated genes identified in the screen are shown with a cutoff for 15-fold enrichment over the genome frequency.

C   Equal amounts of WT cells and indicated single or double mutants in serial dilutions were incubated on agar plates at the indicated temperatures. The synthetic growth defects of *ups4Δ tul1Δ* or *ups4Δ ubx3Δ* double mutants at 26 and 37°C were complemented with plasmids encoding *VPS4* or *TUL1* or *UBX3*. The dashed boxes indicate the non-complemented double mutants.

D   Cartoon presentation of Dsc complex members and evolutionary conserved functional domains with human candidate genes that show the highest homology in the indicated domains.

identified several related E3 ubiquitin ligases (Fig EV1D, Appendix Table S3). Among the best scoring candidates are Rnf122 and Rnf24, two little characterized transmembrane ubiquitin ligases that have been reported to localize to the ER and to the Golgi (Wang *et al*, 2006; Lussier *et al*, 2008).

In budding yeast, the Dsc complex localizes to the Golgi, endosomes, and the limiting membrane of vacuoles (Reggiori & Pelham, 2002; Stewart *et al*, 2011; Li *et al*, 2015b; Yang *et al*, 2018), where it ubiquitinates its substrates. The Dsc substrates are thought to be exclusively degraded inside vacuoles via the ESCRT pathway (Reggiori & Pelham, 2002; Tong *et al*, 2014; Dobzinski *et al*, 2015; Li *et al*, 2015a; Yang *et al*, 2018). It is difficult to rationalize the synthetic genetic interaction of *vps4Δ* with the Dsc complex, if the Dsc complex would only function upstream of the ESCRT machinery. In *Schizosaccharomyces pombe* the Dsc complex also shows negative genetic interaction with the ESCRT machinery. In this fission yeast, the Dsc complex localizes to the Golgi where it mediates the proteolytic activation of the sterol regulatory element-binding protein (*SREBP*), *Sre1,* and thereby controls sterol biosynthesis (Stewart *et al*, 2011; Frost *et al*, 2012; Tong *et al*, 2014). Yet, *SREBP* homologues are absent from the *S. cerevisiae* genome. Hence, unbiased genetic screens (Fig 1A and B) (Costanzo *et al*, 2010, 2016; Stewart *et al*, 2011; Frost *et al*, 2012; Tong *et al*, 2014) collectively suggested that the Dsc complex had unknown functions that were independent of ESCRT-mediated lysosomal degradation. Moreover, these functions appeared to cooperate with the ESCRT pathway to ensure cell survival.

## The Dsc complex ubiquitinates lysine 25 and lysine 33 in the N-terminus of Orm2

To determine the contribution of the Dsc complex to proteostasis independent of vacuolar degradation pathways, we quantitatively profiled global protein turnover of *pep4Δ* single mutants and of *tul1Δ pep4Δ* double mutants (Fig 2A). Vacuolar degradation is severely impaired in *pep4Δ* mutants, and therefore proteasomal degradation will be a key factor for protein turnover. The mutant yeast cells were grown for > 10 generations with heavy (H) lysine ($^{13}C_6\,^{15}N_2$-L-lysine) as the sole lysine source and then diluted in medium containing normal lysine (L, $^{12}C_6\,^{14}N_2$-L-lysine) at the beginning of the experiment (Christiano *et al*, 2014) (Fig 2A). After 180 min, we analyzed the decay of the heavy lysine signal relative to the increase in light lysine in both proteomes by high-resolution mass spectrometry (MS). For the vast majority of the proteins

(2,335 proteins), their turnover was similar in *tul1Δ pep4Δ* double mutants and in *pep4Δ* single mutants. Yet, 76 proteins (Fig 2A and Appendix Table S4) showed at least twofold higher H/L ratios in *tul1Δ pep4Δ* double mutants compared with *pep4Δ* single mutants, indicating that their turnover was reduced. Gene Ontology analysis revealed that these proteins localize most frequently to membrane-bound organelles (plasma membrane and endoplasmic reticulum; Fig EV2A and Appendix Table S5). 32 proteins (42%) had at least one predicted transmembrane domain (Fig 2A, Appendix Table S4). Thus, there seemed to be a considerable number of membrane proteins that are turned over in a Dsc complex-dependent but lysosome-independent manner. Interestingly, potential Dsc candidate substrates were also markedly enriched in proteins functioning in lipid metabolism (nine proteins including eight transmembrane proteins; Fig EV2A; Appendix Tables S4 and S5).

Several lines of evidence pointed to Orm2 as a potentially important Dsc substrate. The analysis of quantitative genetic interaction networks of *tul1Δ* mutants (Costanzo *et al*, 2016) revealed shared genetic interactions with hypomorphic alleles of *lcb1-2* and *lcb2-2* (Fig EV2B). Lcb1/2 are two essential subunits of the serine palmitoyltransferase (SPT). Together with Orm1/2, Tsc3 and Sac1 they form the SPOTS complex (Breslow *et al*, 2010) (Fig 2B). The SPOTS complex localizes to the ER and is essential for the homeostatic regulation of SL biosynthesis because SPT catalyzes the first and rate-limiting step. Tsc3 stimulates SPT activity and *tul1Δ tsc3Δ* double mutants displayed synthetic growth defects (Fig EV2C), consistent with a reported negative genetic interaction of a temperature-sensitive SPT allele (*lcb2-19*) with *tul1Δ* (Fig EV2D) (Li *et al*, 2011). The potential Dsc substrate Orm2 and its paralogue Orm1 inhibit SPT activity (Breslow *et al*, 2010). Orm1/2 are integral membrane proteins with four transmembrane domains and a cytosolic N- and C-terminus (Fig EV2E). Growth defects of *tul1Δ orm1Δ* or *tul1Δ orm2Δ* double mutants were not observed (Fig EV2C). Yet, elevated steady-state proteins levels of Orm2 were detected in deletion mutants of every Dsc complex subunit (*tul1Δ, ubx3Δ, dsc2Δ,* and *dsc3Δ*) (Figs 2C and EV2F). Collectively, these results suggested that the Dsc complex positively regulated the activity of the SPOTS complex.

To characterize how Dsc and SPOTS complexes were mechanistically linked, we analyzed whether components of the SPOTS complex were ubiquitinated by the Dsc complex. Except for Orm2, the protein levels of other SPOTS subunits (Orm1, Sac1, Lcb1/2) were unchanged in *tul1Δ* mutants (Fig 2D), as were *ORM2* mRNA levels (Fig EV2G) (Han *et al*, 2010; Liu *et al*, 2012). To test whether Orm2 was ubiquitinated, we replaced endogenous Orm2 with a

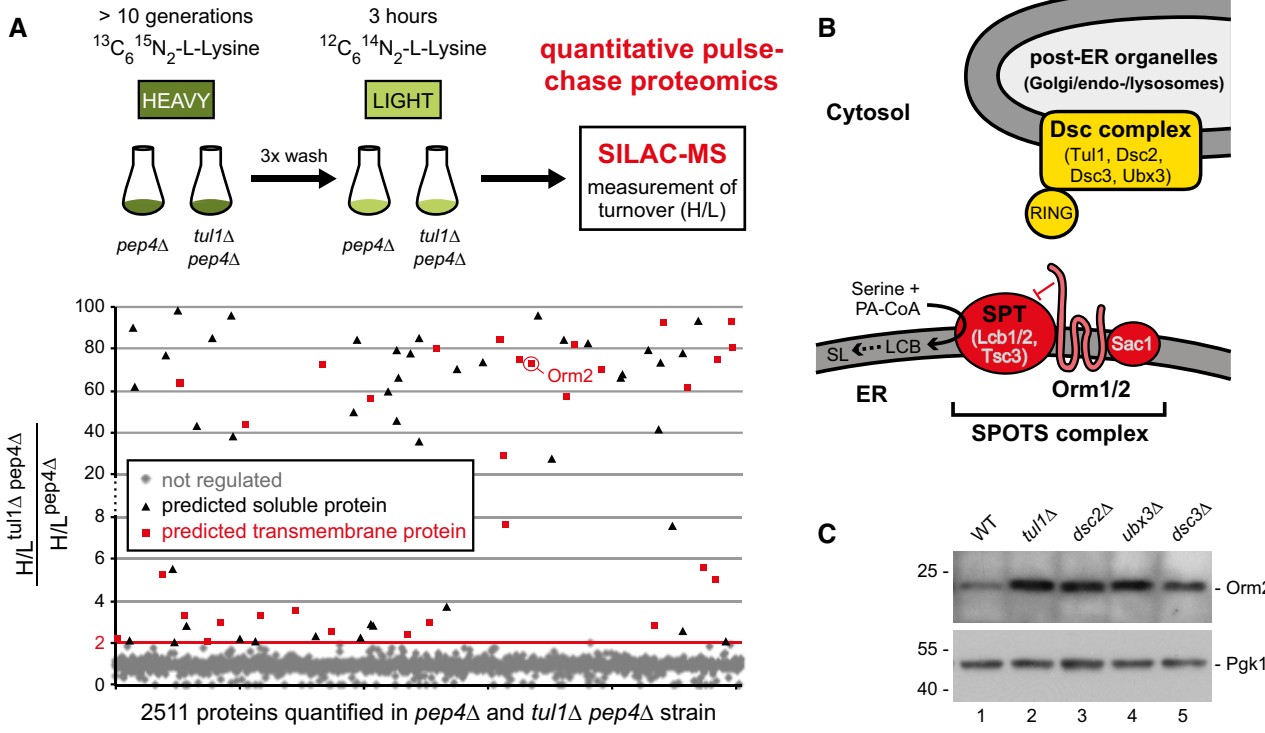

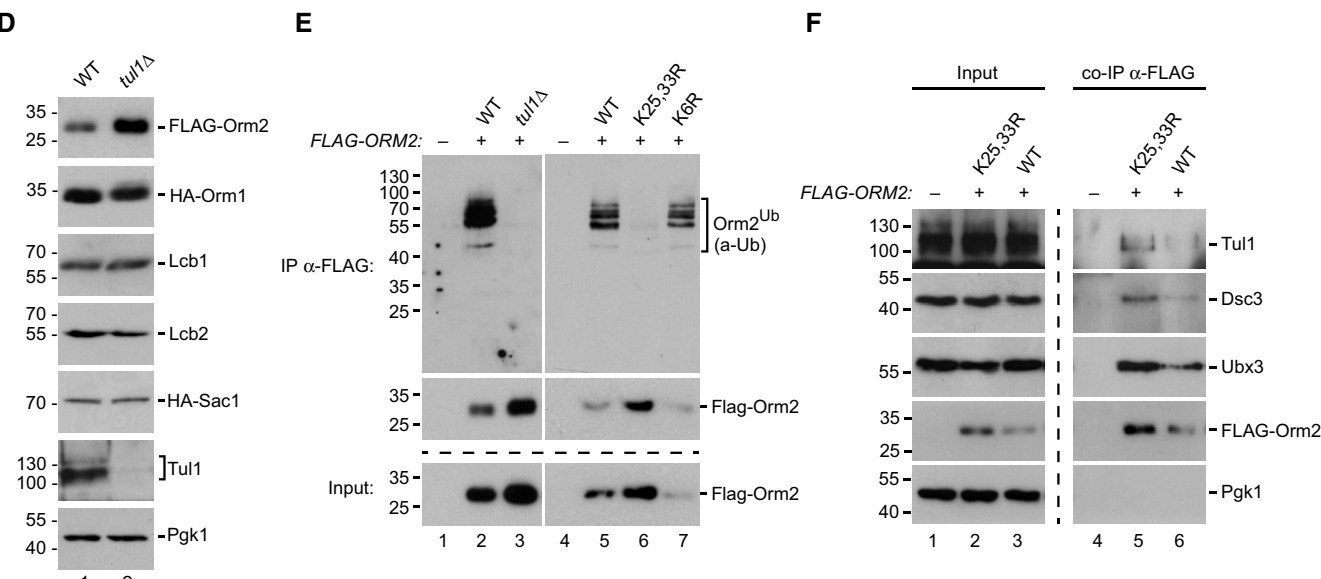

**Figure 2.  Proteome analysis reveals Orm2 as an endogenous Dsc complex substrate.**

See also Fig EV2 and Appendix Tables S4 and S5.

A    Vacuolar proteolysis-deficient *pep4Δ* and *tul1Δ pep4Δ* cells were labeled to saturation with heavy $^{13}C_6^{15}N_2$-L-lysine and then chased for 3 h in the presence of light L-lysine, following quantitative proteome analysis by LC–MS (two biological replicates). To analyze Dsc complex-dependent protein turnover, the ratio of H/L ratios of *tul1Δ pep4Δ* over *pep4Δ* cells of 2511 proteins quantified in both strains was plotted, and proteins with a ratio of H/L ratios > 2 were selected. Red squares: regulated proteins with at least one predicted transmembrane (TM) domain; black triangles: regulated proteins without TM domains.

B    Schematic representation of the Dsc ubiquitin ligase complex (a top hit from the SGA screen) and the SPOTS complex (a putative Dsc substrate). SPT, serine palmitoyltransferase; LCB, long-chain bases; SL, sphingolipids.

C–F   SDS–PAGE and Western blot analysis with the indicated antibodies (C, D) of total yeast lysates from WT cells and the indicated mutants; (E) of input and elution (with FLAG peptide) from denaturing FLAG-Orm2 immunoprecipitations (IP) from WT cells and the indicated mutants. Control cells expressed untagged Orm2; (F) of input and elution (with FLAG peptide) from native FLAG-Orm2 co-immunoprecipitations from WT cells and the indicated mutants. The control strain expresses untagged Orm2.

functional 3xFLAG-tagged version (hereafter FLAG-Orm2) (Breslow *et al*, 2010) and immunoprecipitated it from cell lysates of wild-type (WT) cells and *tul1Δ* mutants under denaturing conditions. Ubiquitinated Orm2 was recovered from WT cells but not from *tul1Δ* cells (Fig 2E). FLAG-Orm2 migrated on an SDS–PAGE at around 30 kD. Ubiquitinated Orm2 was detected at about 55–100 kD with frequent laddering suggesting poly-ubiquitination (Figs 2E and 5B–D). To identify the potential ubiquitination sites in Orm2, we mutated the lysine residues predicted to face the cytosol to arginine (Fig EV2E). Orm2-K25,33R (but not Orm2-K6R or K25R and K33R single mutants) was expressed at higher levels compared with WT and was no longer efficiently ubiquitinated (Figs 2E and EV2H). Under non-denaturing lysis conditions, Orm2 co-immunoprecipitated Tul1, Dsc3, and Ubx3 and the Orm2-K25,33R mutant co-immunoprecipitated higher levels of Dsc complex subunits (Fig 2F).

These results showed that the Dsc complex interacted with Orm2 and mediated its ubiquitination at two N-terminal lysine residues (K25 and K33). Thus, Orm2 is likely an endogenous substrate of the Dsc complex.

## Ubiquitination of Orm2 prevents its accumulation in Golgi and endosomes

The majority of GFP-Orm2 was detected at the ER by live-cell, wide-field, and confocal fluorescence microscopy in WT cells as described earlier (Breslow *et al*, 2010; Figs 3A and EV3A). In *tul1Δ* mutants, GFP-Orm2 was detected at the ER and it additionally accumulated in discrete objects outside the ER (in 99% of 187 cells, Figs 3A and EV3A). These post-ER compartments co-localized partially with markers for the Golgi (Sec7-mCherry) (Day *et al*, 2018) but more frequently with Vps4-mCherry on endosomes/MVBs (Adell *et al*, 2017) (Fig 3B). GFP-Orm2-K25,33R was also detected at the ER and on post-ER compartments (in 72% of 269 cells, Fig 3A, arrowheads). Hence, non-ubiquitinated Orm2 accumulated at the ER and in post-ER compartments.

The Dsc complex uses two trafficking adaptors, Gld1 or Vld1, to target to different post-ER compartments (Yang *et al*, 2018). Gld1 binding sorts the Dsc complex to the Golgi and to endosomes, while Vld1 diverts the Dsc complex from the Golgi via the AP-3 pathway directly to the limiting membrane of the vacuole. In *gld1Δ* mutants, the Dsc complex is exclusively detected on the limiting membrane of vacuoles (Yang *et al*, 2018). Disrupting Gld1 resulted in GFP-Orm2 accumulation at the ER and post-ER compartments (Fig 3A, fully penetrant phenotype, *n* = 153 cells), similar to *tul1Δ*. The deletion of Vld1 did not alter the ER localization of GFP-Orm2 (Fig 3A). In *vld1Δ* mutants, the Dsc complex is no longer detected on the limiting membrane of the vacuole, but still localizes to Golgi and endosomes (Yang *et al*, 2018). These results suggested that the Gld1-Dsc complex prevented the accumulation of Orm2 on endosomes and Golgi.

The Dsc complex also prevented the aberrant accumulation of GFP-Orm2 in class E compartments in ESCRT mutants. When the Dsc complex was active in ESCRT mutants (*vps4Δ* cells), only *bona fide* ESCRT cargoes (e.g., mCherry-Cps1) were detected in class E compartments, whereas GFP-Orm2 was mainly detected at the ER. However, upon disruption of the Dsc complex in *tul1Δ vps4Δ* double mutants, GFP-Orm2 accumulated additionally on class E compartments together with mCherry-Cps1 (Fig 3C). Consistently,

the protein levels of Orm2 were elevated in *tul1Δ vps4Δ* double mutants compared with *vps4Δ* single mutants (Fig EV3B). The deletion of *ORM2* hardly restored the growth of *tul1Δ vps4Δ* double mutants (Fig 3C). The *vps4Δ tul1Δ orm2Δ* triple mutants still grew poorly when compared to *vps4Δ orm2Δ* and *tul1Δ orm2Δ* double mutants (Fig 3C). It seemed that, in addition to Orm2, other critical substrates of the Dsc complex were degraded by ESCRT-independent pathways.

All together, these observations indicate that the Gld1-Dsc complex selectively targets the membrane protein Orm2 from the Golgi and endosomes for degradation in an ESCRT-independent manner.

## Ubiquitinated Orm2 is degraded by proteasomes

To characterize how ubiquitinated Orm2 was degraded, we blocked protein translation and followed Orm2 protein levels. In WT cells, 70–80% of Orm2 was degraded after 3 h (Fig 4A and B), consistent with the analysis from global protein turnover (Fig 2A). In mutants of all four subunits of the Dsc complex and in *gld1Δ* (but not in *vld1Δ*) mutants, the degradation of Orm2 was blocked (Figs 4A and B, and EV4A). Likewise, the degradation of Orm2-K25,33R was severely impaired (Fig 4A and B). Orm2 degradation was independent of vacuolar degradation (*pep4Δ* mutants; Fig 4A and B) or autophagy (*atg8Δ* mutants; Fig EV4B), and we never observed GFP-Orm2 inside vacuoles (Figs 3A and EV4C).

In contrast, inhibition of the proteasome in *cim3-1* mutants (a temperature-sensitive mutant of the 19S particle; Ghislain *et al*, 1993) or with MG-132 (in *pdr5Δ* mutants to prevent MG-132 efflux) strongly impaired Orm2 degradation (Figs 4A and C and EV4D). Proteasomal degradation of membrane proteins at the ER typically requires ERAD where the E3 ubiquitin ligases, Hrd1 and Doa10, and their associated proteins retro-translocate and poly-ubiquitinate membrane proteins. At the INM, membrane protein degradation is mediated by the Asi complex. For ERAD and Asi-ERAD, the AAA-ATPase Cdc48/VCP then extracts poly-ubiquitinated proteins from membranes prior to proteasomal degradation (Bays *et al*, 2001; Ye *et al*, 2001; Jarosch *et al*, 2002; Bodnar & Rapoport, 2017). Neither Hrd1 and Doa10 nor Asi1 were required for Orm2 degradation (Figs 4A and B, and EV4B). Yet, the inactivation of Cdc48 (temperature-sensitive allele *cdc48-3*; Madeo *et al*, 1997) impaired Orm2 degradation (Fig 4A and C) and caused accumulation of GFP-Orm2 in post-ER compartments (Fig 4D), similar to *tul1Δ* mutants (Fig 3A). To test whether Orm2 degradation required its transport from the ER to the Golgi, we blocked COP-II function using the temperature-sensitive *sec13-4* mutant. At the non-permissive temperature, Orm2 degradation was strongly impaired in *sec13-4* mutants (Fig 4C). Consistently, 120 min after the shift to the non-permissive temperature, total Orm2 protein levels increased in *sec13-4* (Fig EV4E) mutants and GFP-Orm2 accumulated at the ER (Fig EV4F, upper panel). Hence, ER export was a prerequisite for the degradation of Orm2. Moreover, Orm2 became depleted from endosomes and Golgi in *tul1Δ* cells upon COP-II inhibition and instead was only detected at the ER (Fig 4E), while Vps4-eGFP-positive endosomes were still detected (Fig EV4F, lower panel). It seemed that accumulating Orm2 could be retrieved back into the ER. Orm1 shares 72% amino acid identity with Orm2, yet it was more stable and not degraded via the Dsc complex (Fig EV4G) and

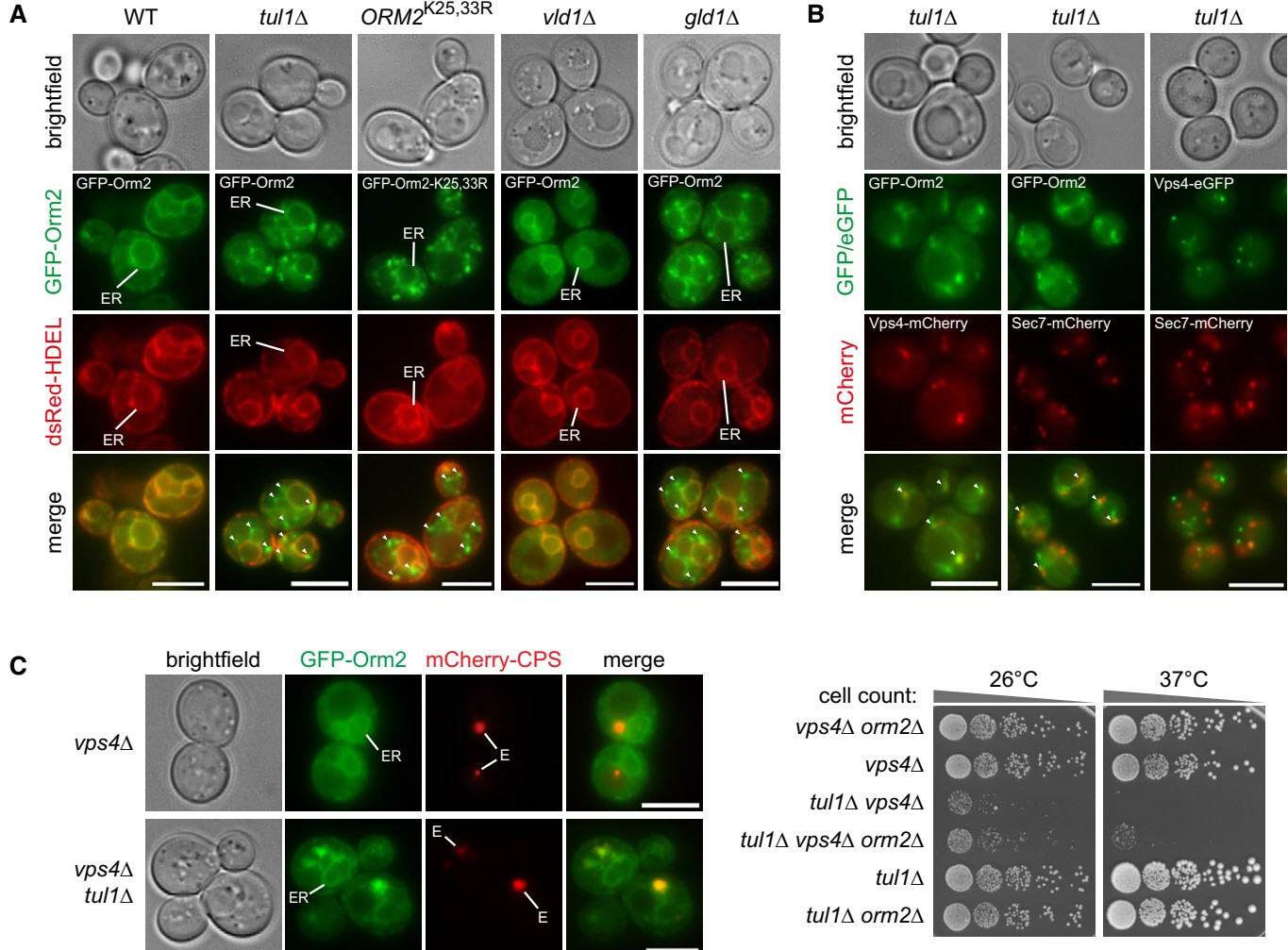

**Figure 3. Non-ubiquitinated Orm2 accumulates in Golgi and endosome structures.**

See also Fig EV3.

A   Epifluorescence and phase-contrast microscopy of living WT yeast cells and the indicated mutants expressing GFP-Orm2 or GFP-Orm2-K25,33R (green) with dsRed-HDEL (red). Arrowheads point to GFP-Orm2 accumulations outside the endoplasmic reticulum (ER).

B   Epifluorescence and phase-contrast microscopy of *tul1Δ* mutants expressing GFP-Orm2 (green) with Vps4-mCherry (red, endosomes) or with Sec7-mCherry (red, Golgi) or Vps4-eGFP (green) with Sec7-mCherry.

C   Left panel: epifluorescence and phase-contrast microscopy of the indicated mutants expressing GFP-Orm2 (green) with mCherry-Cps1 (red, MVB cargo). E indicates class E compartments; right panel: Equal amounts of the indicated mutants in serial dilutions were incubated on agar plates at the indicated temperatures.

Data information: scale bars = 5 μm.

GFP-Orm1 was only detected at the ER in *tul1Δ* mutants (Fig EV4H).

These results demonstrated that the selective ubiquitination of Orm2 by the Dsc complex was a prerequisite for proteasomal degradation, which was independent of the ERAD ubiquitin ligase complexes and the Asi complex but dependent on Cdc48 activity and ER export to Golgi and endosomes.

## Cdc48/VCP extracts ubiquitinated Orm2 for proteasomal degradation

To better define the Orm2 degradation process, we tested whether ubiquitinated Orm2 became soluble prior to proteasomal

degradation or whether it was directly degraded while associated with membranes (Mayer *et al*, 1998; Lee *et al*, 2004; Smith *et al*, 2016). Therefore, cells were lysed (without detergent) after proteasome inhibition and the insoluble membrane fraction (P100) was separated from the soluble cytosolic fraction (S100) by centrifugation at 100,000 × *g* (Neal *et al*, 2017, 2018) (Fig 5A). Orm2 was immunoprecipitated from these fractions after denaturing and its ubiquitination was analyzed. Upon inhibition of proteasomal degradation, ubiquitinated Orm2 accumulated predominantly in the insoluble membrane fraction (P100) in WT cells (Fig 5B). In the cytosolic fraction (S100) of WT cells, a small fraction of ubiquitinated Orm2 was detected. Soluble non-ubiquitinated Orm2 was not detected. In *tul1Δ* mutants, ubiquitinated Orm2 was absent from

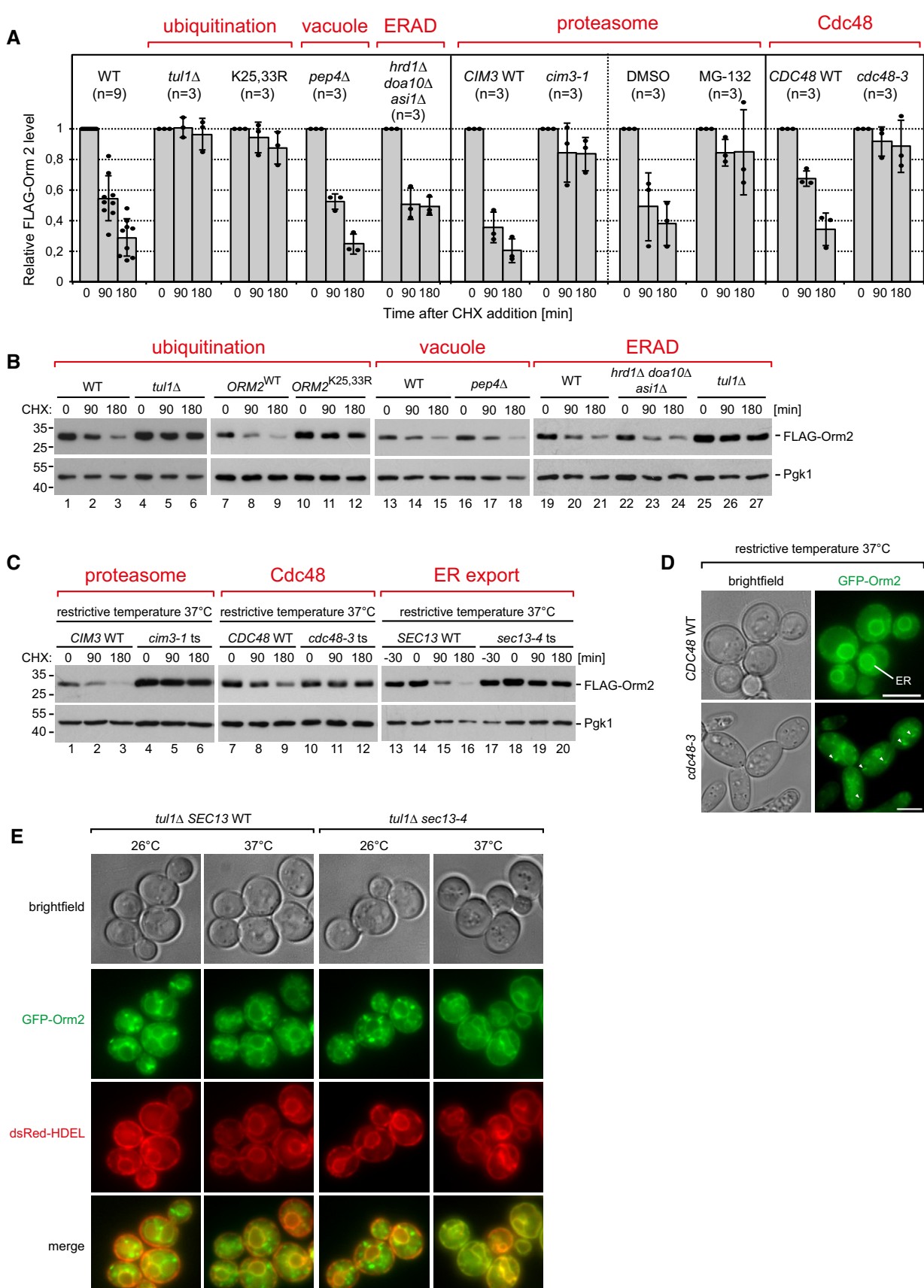

Figure 4.

◀

**Figure 4.  The Dsc complex mediates degradation of Orm2 at cytosolic proteasomes.**

See also Fig EV4.

A–C  WT cells and the indicated mutants were left untreated (0 min) or were treated with cycloheximide (CHX) to block protein synthesis for 90 and 180 min at 26°C or at the non-permissive temperature (37°C), where indicated. Total cell lysates were analyzed by SDS–PAGE and Western blotting with the indicated antibodies. (A) Graphs display the FLAG-Orm2 protein levels determined by densitometric quantification of Western blots from cell lysates of WT cells and the indicated mutants 0, 90, and 180 min after the addition of CHX. Each experiment was repeated at least three times, FLAG-Orm2 levels were normalized to Pgk1 loading controls, and each time point was related to $t = 0$ min (set to 1). Data are presented as mean $\pm$ standard deviation. (C) *SEC13* WT and *sec13-4* cells were pre-incubated for 30 min at 37°C before CHX was added.

D  Epifluorescence and phase-contrast microscopy of living WT cells and *cdc48*-3 mutants expressing GFP-Orm2 incubated for 5 h at the non-permissive temperature (37°C).

E  Epifluorescence and phase-contrast microscopy of living *tul1Δ* and *tul1Δ sec13-4* mutants expressing GFP-Orm2 (green) and dsRed-HDEL (red) incubated at 26°C or shifted for 90 min to the non-permissive temperature (37°C).

Data information: scale bars 5 μm.

membrane or soluble fractions. In *cdc48-3* cells shifted to the restrictive temperature, ubiquitinated Orm2 accumulated in the membrane fraction already without proteasome inhibition but was never released into the soluble fraction (Fig 5C). To analyze how ubiquitination changed the association of Orm2 with membranes, sodium carbonate ($Na_2CO_3$) extraction experiments were performed. $Na_2CO_3$ preferentially extracted ubiquitinated Orm2. Even under harsh conditions (pH 12–12.5), only little non-ubiquitinated Orm2 was solubilized (Fig 5D, lanes 5 and 6), suggesting that ubiquitination facilitated the extraction of Orm2 from membranes.

It seemed that ubiquitinated Orm2 was extracted from post-ER membranes in a Cdc48-dependent process. As only little ubiquitinated Orm2 accumulated in the cytosol, membrane extraction of ubiquitinated Orm2 by Cdc48 was likely tightly coupled to proteasomal activity and degradation. These results suggested that Orm2 is a substrate of a novel pathway for the selective proteasomal degradation of membrane proteins from Golgi and endosomes, which we term endosome and Golgi-associated degradation (EGAD) (Fig 5E).

**EGAD-dependent degradation of Orm2 maintains sphingolipid homeostasis**

Next, we tested how the EGAD-dependent degradation of Orm2 contributed to the homeostatic regulation of sphingolipid metabolism. Orm2 and its paralogue Orm1 inhibit the enzymatic activity of the SPT. SPT condenses serine and fatty acids (palmitoyl-CoA) into long-chain bases (LCB), which represent the first and rate-limiting metabolite in SL biosynthesis. LCBs are immediately consumed to form ceramides and complex SL. Thus, overexpression of Orm1/2 and the ensuing inhibition of SPT were found to reduce the overall levels of ceramides and of most complex SLs (Breslow *et al*, 2010; Han *et al*, 2010; Liu *et al*, 2012; Siow & Wattenberg, 2012). Indeed, mass-spectrometric analysis of lipid extracts from *tul1Δ* and Orm2-K25,33R mutants, which have higher Orm2 protein levels, showed reduced ceramide levels (Fig 6A). Furthermore, lipid extracts from WT cells and *tul1Δ* mutants were analyzed by thin-layer chromatography. *tul1Δ* mutants showed reduced SPT-dependent incorporation of [$^3$H]-serine into complex SL—mainly inositolphosphorylceramide (IPC) and mannosylinositolphosphorylceramide (MIPC) (Fig 6B).

Of note, accumulating Orm2-K25,33R co-immunoprecipitated more efficiently the ER-resident catalytic SPT subunits Lcb1/2 compared with WT Orm2 (Fig 6C). It seemed that EGAD-dependent degradation of Orm2 restricted its interaction with the SPT. These results explain how Orm2 accumulation repressed the synthesis of SL in cells with a defective EGAD pathway.

Orm1/2 function is controlled by the target of rapamycin complex 2 (TORC2) and by its downstream AGC-family protein kinases Ypk1/2 (Aronova *et al*, 2008; Breslow *et al*, 2010; Roelants *et al*, 2011). Plasma membrane stress and decreasing SL levels activate TORC2, which in turn phosphorylates and fully activates Ypk1/2 (Aronova *et al*, 2008; Berchtold *et al*, 2012; Niles *et al*, 2012; Riggi *et al*, 2018). Activated Ypk1/2 phosphorylate Orm1/2 and phosphorylated Orm1/2 no longer inhibit SPT activity, which de-represses SL synthesis and restores membrane homeostasis.

As expected, the defects in sphingolipid homeostasis in *tul1Δ* mutants or in cells expressing Orm2-K25,33R correlated with increased Ypk1/2-mediated phosphorylation of Orm2 and Orm1 (Figs 6D and E, and EV5A and B). This was accompanied by modest activation of Ypk1 by TORC2 (Ypk1 pT662) in cells expressing Orm2-K25,33R (Fig 6D) and in *tul1Δ* mutants (Fig 6E, lane 3, 4). Cells responded to the accumulation of Orm2 by phosphorylating it via TORC2-Ypk1 signaling. Yet, Orm2 phosphorylation was no longer sufficient to restore SL homeostasis when its EGAD-dependent degradation was blocked (Fig 6F). Thus, EGAD-dependent degradation of Orm2 essentially contributed to the regulation of SL metabolism.

**Phosphorylation of Orm2 by TORC2-Ypk1 signaling promotes ER export and EGAD-dependent degradation**

We next tested whether and how Orm2 phosphorylation contributed to its EGAD-dependent degradation. Therefore, we first acutely inhibited TORC2 (Gaubitz *et al*, 2015). Acute inhibition of TORC2 impaired the degradation of Orm2 (Fig 7A and B). Downstream of TORC2, Ypk1/2 phosphorylates Orm2 on serine residues 46, 47, and 48 (Roelants *et al*, 2011). Orm2-3A (S46, 47, 48A) was not phosphorylated (Fig 6E) and no longer efficiently degraded at least within 180 min (Fig 7A and B). Moreover, Orm2-3A was also not efficiently ubiquitinated (Fig 7C, lane 3). At best residual, ubiquitination was detected. These results showed that TORC2 signaling and Ypk1/2-dependent phosphorylation of Orm2 promoted its EGAD-dependent degradation. In contrast, Npr1 phosphorylation of Orm2, which stimulates complex sphingolipid synthesis (Shimobayashi *et al*, 2013), was dispensable for Orm2 degradation. The respective Orm2-7A (S9, 15, 22, 29, 31A, T18, 36A) was still degraded (Fig 7A and B).

Of note, GFP-Orm2-3A accumulated almost exclusively in the ER in *tul1Δ* (fully penetrant phenotype, $n = 78$ cells) and *cdc48-3* mutants (Figs 7D and EV5C). It seemed that Orm2 phosphorylation triggered its export from the ER. Consistently, Orm2-3A

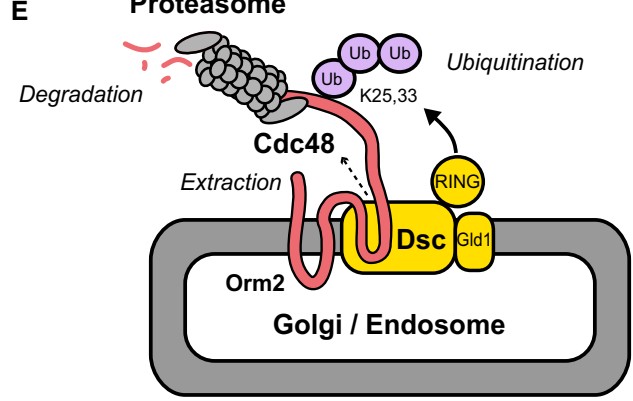

**Figure 5.**

**Figure 5.  Orm2 is retro-translocated by the Dsc complex and Cdc48 for proteasomal degradation.**

A    WT (*pdr5Δ*) cells expressing FLAG-Orm2 and Sec61-GFP (integral ER membrane protein) treated with MG-132 were fractionated into insoluble membrane (P100) and soluble cytoplasmic (S100) fractions and analyzed by SDS–PAGE and Western blot with the indicated antibodies.

B, C    WT cells, *tul1Δ*, and *cdc48-3* mutants (all in a *pdr5Δ* strain background) were left untreated or treated with the proteasome inhibitor (MG-132) as indicated and subsequently subjected to subcellular fractionation as in (A). S100 and P100 fractions were subjected to denaturing FLAG-Orm2 immunoprecipitations (IP). Input and eluted fractions (with FLAG peptide) were analyzed by SDS–PAGE and Western blotting with the indicated antibodies.

D    P100 fractions from MG-132-treated WT (*pdr5Δ*) cells expressing FLAG-Orm2 and Sec61-GFP were extracted with $Na_2CO_3$ solution at the indicated pH and subsequently fractionated by $100,000 \times g$ centrifugation into soluble (S) and insoluble membrane (M) fractions, subjected to denaturing FLAG-immunoprecipitations and analyzed by Western blotting with the indicated antibodies. The asterisk indicates reactivity of the secondary antibody with IgG heavy chains.

E    Schematic representation of Orm2 degradation by the EGAD pathway.

accumulation in the ER hyper-activated TORC2-dependent phosphorylation of Ypk1, similar to inhibiting SPT activity with myriocin (Fig 6E, lanes 5–9). Due to the impaired degradation of Orm2-3A, its protein levels were always higher compared with WT cells (Figs 6E and EV5D). Yet, the protein levels of Orm2-3A still increased moderately in *tul1Δ* mutants (Figs 6E and EV5D). Thus, it seemed possible that a small fraction of Orm2-3A escaped from the ER to Golgi and endosomes where it was degraded by EGAD.

In contrast to the Orm2-3A mutant, the phospho-mimetic Orm2-3D (S46, 47, 48D) protein was constitutively exported from the ER and targeted for EGAD-dependent degradation. Orm2-3D protein levels were severely decreased (Fig 6E, compare lane 1 and lane 3) but accumulated in *tul1Δ* mutants (Fig 6E, lane 2). Unlike Orm2-3A, Orm2-3D was still efficiently ubiquitinated (Fig 7C, lane 4). In *tul1Δ* mutants, GFP-Orm2-3D was no longer detected at the ER but localized almost exclusively to post-ER compartments (in 91% of 80 cells) (Fig 7D).

Based on these results, we hypothesized that Orm2-3D, which is constitutively exported from the ER and degraded (Figs 6E and 7D), should only poorly interact with the ER-resident SPT subunits Lcb1/2. In contrast, Orm2-3A, which is trapped at the ER and barely degraded (Figs 6E and 7D), should interact with SPT, similar to Orm2-K25,33R. To directly test this prediction, we introduced K25,33R mutations to prevent the constitutive degradation of Orm2-3D. The protein levels of Orm2-K25,33R-3D increased and were comparable to those of Orm2-K25,33R-3A or Orm2-K25,33R (Fig 7E). Orm2-K25,33R-3D hardly co-immunoprecipitated Lcb1/2 (Fig 7E). Yet, it interacted with the Dsc complex subunits on Golgi and endosomes. Conversely, Orm2-K25,33R-3A co-immunoprecipitated efficiently Lcb1/2, but only poorly interacted with the subunits of the Dsc complex (Fig 7E). Orm2-K25,33R co-immunoprecipitated Lcb1/2 in amounts comparable to Orm2-K25,33R-3A as well as the subunits of the Dsc complex. Consistent with these results, GFP-Orm2-K25,33R-3D was mainly detected in post-ER compartments, GFP-Orm2-K25,33R-3A was exclusively detected at the ER (Fig EV5E), and GFP-Orm2-K25,33R localized to the ER and in post-ER compartments (Fig 3A).

These results confirm a key prediction and show that the failure to export Orm2 from the ER and the failure to degrade Orm2 increase the interaction with SPT at the ER and therefore should both repress sphingolipid biosynthesis. Indeed, the synthesis of LCB and ceramide species was repressed to similar extents in *tul1Δ* mutants and in cells expressing Orm2-3A and Orm2-K25,33R (Figs 7F and 6A). In contrast, Orm2-3D disproportionately de-repressed sphingolipid synthesis with levels of LCB and ceramide species above those of WT cells (Fig 7F).

Based on these results, we conclude that TORC2-Ypk1-dependent phosphorylation of Orm2 triggers ER export to Golgi and endosomes. There, Orm2 is ubiquitinated by the Dsc complex, extracted from membranes by Cdc48, and degraded by proteasomes. EGAD-dependent degradation of Orm2 restricts the retrieval of Orm2 back to the ER, limits the re-association with the SPT, and thereby promotes controlled de-repression of sphingolipid synthesis. Thus, sphingolipid homeostasis in yeast requires Orm2 ER export and its subsequent degradation by EGAD.

## Discussion

The model (Fig 7G) provides the conceptual and mechanistic framework for EGAD. EGAD is a pathway for the selective degradation of integral membrane proteins by proteasomes that operates in addition to ESCRT and ERAD systems. The current model for EGAD is based on the characterization of its substrate Orm2, which is exclusively degraded by EGAD. In the absence of a functional EGAD pathway, Orm2 accumulates at the ER and on post-ER membranes. For reasons currently unclear, it cannot be efficiently degraded via the ESCRT machinery or ERAD. Thus, some membrane proteins in eukaryotic cells appear to have characteristics that make them preferential EGAD substrates and poor substrates for other degradation systems. Indeed, our genetic experiments indicate that Orm2 is not the only EGAD substrate. Also, quantitative profiling of proteome turnover identified several other membrane proteins that could be candidate substrates of the Dsc complex and EGAD-dependent degradation. Like Orm2, some of these membrane proteins are involved in the regulation of lipid metabolism. Future studies will clarify, if they are direct clients of EGAD-mediated degradation. We speculate that eukaryotic cells employ EGAD to prevent the accumulation of membrane proteins that are otherwise poor substrates for ESCRT or ERAD pathways.

EGAD-dependent degradation of Orm2 is a multi-step process (Fig 7G) and requires (i) regulated export of Orm2 from the ER; (ii) Dsc complex-dependent recognition of Orm2 at the Golgi and on endosomes; (iii) poly-ubiquitination; and (iv) Cdc48/VCP-dependent membrane extraction of ubiquitinated Orm2 in tight coordination with (v) its proteasomal degradation. Failure to degrade Orm2 causes its accumulation in the ER and in post-ER compartments. There is evidence that EGAD systems operate in other organisms. In fission yeast, failure to mature the SREBP precursor (Sre1) in *rbd2Δ* mutants results in its proteasome-dependent and Dsc complex-dependent degradation (Hwang *et al*, 2016). While there is no direct evidence that ubiquitinated immature Sre1 is extracted from

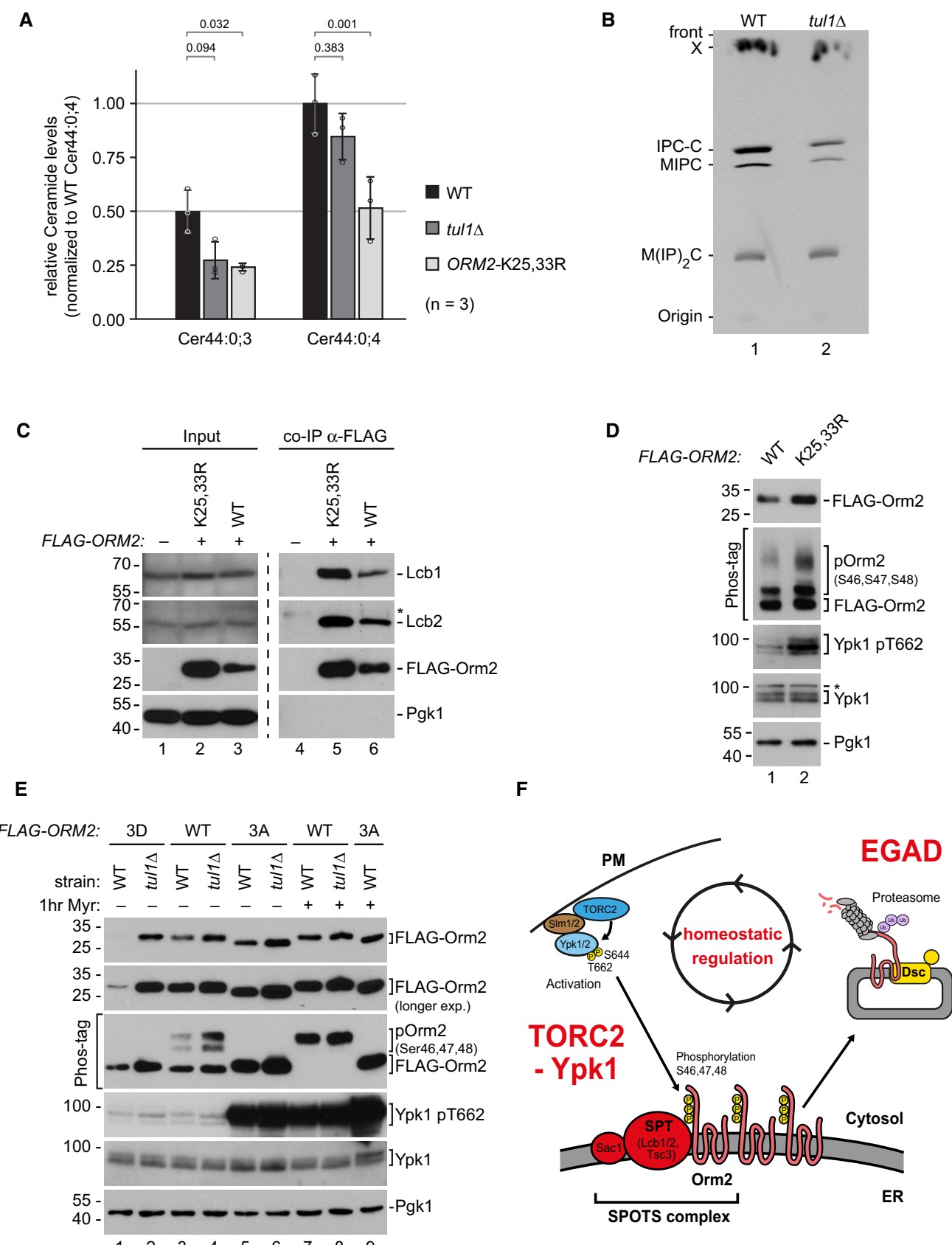

**Figure 6.**

**Figure 6.   EGAD-degradation of Orm2 contributes to sphingolipid homeostasis.**

See also Fig EV5.

A   The levels of two major ceramide (Cer) species in lipid extracts from WT (black), *tul1Δ* (dark gray), or Orm2-K25,33R mutants (light gray) were measured using LC-MS and quantified using spiked non-yeast ceramides as internal standard. Data are normalized to Cer44:0;4 levels of WT cells (set to 1) and presented as mean ± standard deviation from three independent experiments. Pairwise statistical significance was assessed by Student's *t*-test. *P* values are given above the respective bars.

B   Thin-layer chromatography of sphingolipid extracts from [$^3$H]serine radiolabelled WT cells and *tul1Δ* mutants was analyzed by autoradiography. IPC, inositolphosphorylceramide; MIPC, mannosylinositolphosphorylceramide; MIP$_2$C, mannosyldiinositolphosphorylceramide. X indicates an unknown lipid species that is insensitive to myriocin treatment.

C   SDS–PAGE and Western blot analysis with the indicated antibodies of input and elution (with FLAG peptide) from native anti-FLAG co-immunoprecipitations from cells expressing FLAG-Orm2-WT or FLAG-Orm2-K25,33R. Control cells expressed untagged Orm2 WT.

D, E   SDS–PAGE or Phos-tag SDS–PAGE and Western blot analysis with the indicated antibodies of total cell lysates (D) from WT cells expressing FLAG-Orm2 or FLAG-Orm2-K25,33R; (E) from WT cells and *tul1Δ* mutants expressing FLAG-Orm2, FLAG-Orm2-3A, or FLAG-Orm2-3D with or without myriocin treatment (1.5 µM) for 1 h.

F   Schematic representation of the regulation of SPT activity via TORC2-Ypk1/2-dependent phosphorylation of Orm proteins and EGAD-dependent degradation of Orm2.

Data information: (C, D) The asterisks label unspecific cross-reactions of the respective antibodies.

membranes, we propose that its proteasomal degradation follows the EGAD pathway.

It is currently unclear how substrate recognition for EGAD is achieved. We propose that subunits of the Dsc complex and/or other unknown components of EGAD systems execute this quality control function. Somehow, the EGAD machinery is capable of detecting orphaned (Orm2) and possibly immature (Sre1) membrane proteins in post-ER compartments and select them for proteasomal degradation. A quality control function of Tul1 at Golgi, endosomes, and the vacuole was previously shown for proteins with polar transmembrane domains. Interestingly, these proteins were not degraded via EGAD, but sorted by the ESCRT machinery for lysosomal degradation (Reggiori & Pelham, 2001; Valdez-Taubas & Pelham, 2005; Dobzinski *et al*, 2015; Li *et al*, 2015a) (Fig 7G). Hence, substrate ubiquitination by the Dsc complex alone does not decide which proteins become ESCRT or EGAD substrates. We suggest that intrinsic properties of membrane proteins that modulate their interaction with the Dsc complex discriminate EGAD substrates from ESCRT substrates. These may include the length, number, and polarity of transmembrane domains (as in ERAD-M) or specific recognition patterns in the luminal domains (as in ERAD-L) such as specific glycosylation or the lack of glycosylation.

Failure of EGAD disrupted SL homeostasis in yeast due to the aberrant accumulation of Orm2 at the ER, Golgi, and endosomes. We speculate that exit from the ER, ubiquitination, membrane extraction, and degradation are functionally linked to prevent retrieval of Orm2 and efficiently sequester it from the ER-resident SPT for homeostatic SL regulation. To a certain degree, this pathway conceptually parallels the regulation of sterol biosynthesis by the Dsc complex in fission yeast (Stewart *et al*, 2011) and in human cells via regulated ER export and post-ER proteolytic maturation of SREBP (Brown *et al*, 2018). Intriguingly, human macrophages fed with excess free cholesterol increased SPT activity by exporting ORMDL1 and 3 from the ER. Upon ER export, ORMDL1/3 accumulated in cytoplasmic punctate structures (Wang *et al*, 2015). How ER exit of ORMDL1/3 and subsequent steps are controlled is not understood. In yeast, TORC2-Ypk1-dependent phosphorylation of Orm2 initiates ER exit. Yet, the regulatory N-terminus of Orm2 including phosphorylation and ubiquitination sites is absent in ORMDL1/3, suggesting that the initial signal that triggers ORMDL1/3 ER exit must differ between human cells and yeast. Interestingly, high-throughput studies identified a C-terminal ubiquitin

modification (K152) in ORMDL1/3 (http://ptmfunc.com) (Beltrao *et al*, 2012). Moreover, the cytoplasmic ORMDL accumulations were positive for the ubiquitin-binding autophagy receptor p62, which was required to clear the cytosolic ORMDL aggregates via selective autophagy. Therefore, it is not unlikely that ubiquitinated ORMDL proteins are extracted from membranes in a process that is similar to EGAD. Given that elevated ORMDL3 protein levels are associated with diabetes, ulcerative colitis, Crohn's disease, and asthma (Moffatt *et al*, 2007; Barrett *et al*, 2008; Breslow *et al*, 2010; McGovern *et al*, 2010), the sequestration of ORM family proteins from the ER and their subsequent ubiquitin-dependent degradation likely has pathophysiological implications.

Direct orthologs of Tul1, the E3-ligase of the Dsc complex, are widespread in evolution, but appear to be absent in the metazoan clade. It is thus difficult to find obvious candidates in the human genome. Several of the 49 different membrane-spanning RING finger E3 ubiquitin ligases in humans (Nakamura, 2011) are potential candidates (e.g., Rnf122 and Rnf24) to ubiquitinate membrane proteins for EGAD at the Golgi, endosomes, and lysosomes. Two other core subunits of the Dsc complex (Ubx3 and Dsc2) have discernible orthologues in the human genome (UBAC2 and FAF2/UBXD8), which function together in the regulation of lipid droplet turnover (Olzmann *et al*, 2013) and moreover associate with ERAD complexes (Mueller *et al*, 2008). Good candidates for adopting Dsc3 function in human cells are TMUB1/2, which also comprise an ubiquitin-like domain and two carboxy-terminal transmembrane regions. Similar to UBAC2 and FAF2, TMUB1 has been shown to be part of the gp78 ERAD ubiquitin ligase complex (Jo *et al*, 2011). Thus, it is conceivable that UBAC2, FAF2, and TMUB1/2 associate with different transmembrane ubiquitin ligase complexes to ensure membrane protein homeostasis at the ER (ERAD-like) and on post-ER compartments (EGAD-like). Moreover, a conceptually similar chloroplast-associated protein degradation (CHLORAD) was recently described (Ling *et al*, 2019).

Hence, we speculate that membrane-embedded multi-subunit ubiquitin ligase complexes on different organelles mediate the selective proteasomal degradation of membrane proteins. Thereby, these "organelle-associated degradation" pathways could be essential to maintain organelle homeostasis. The characterization of the EGAD machinery in yeast now allows to systematically address the underlying molecular mechanisms and thereby opens new perspectives for a more complete understanding of proteostasis in eukaryotic cells.

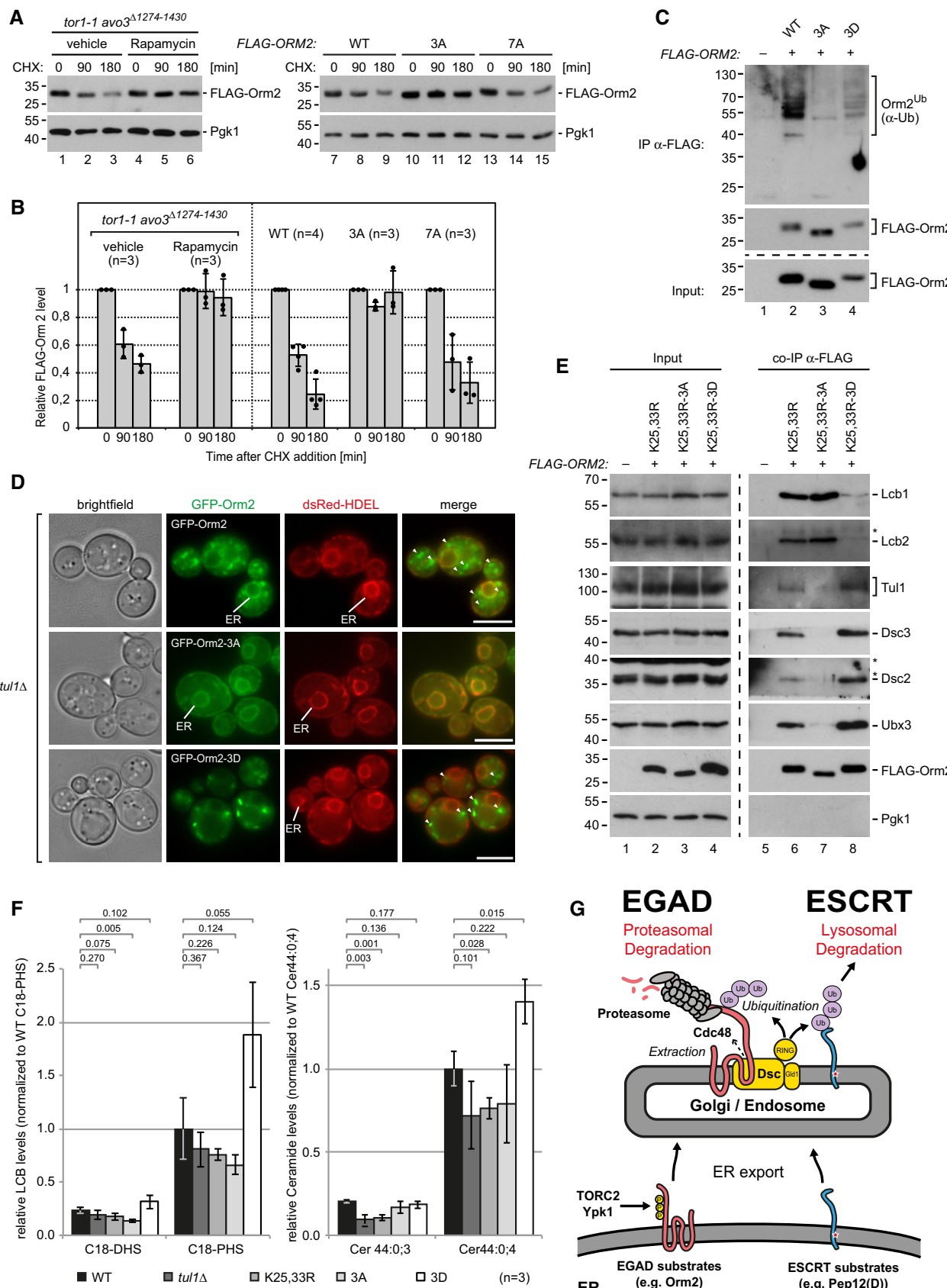

**Figure 7.**

◀

Figure 7.  ER exit of Orm2 is regulated by TORC2-Ypk1 phosphorylation and essential for Orm2 degradation.

See also Fig EV5.

A   Left panel: In *tor1-1 avo3*$^{\Delta 1274-1430}$ cells, rapamycin specifically inhibits TORC2 but not TORC1 (Gaubitz *et al*, 2015). *tor1-1 avo3*$^{\Delta 1274-1430}$ cells were treated with cycloheximide (CHX) and rapamycin or vehicle (methanol) for the indicated times. Right panel: WT or Orm2 mutant cells were treated with CHX for the indicated times. The protein levels of FLAG-Orm2 or of the indicated FLAG-Orm2 mutants from total cell lysates were analyzed by SDS–PAGE and Western blotting with the indicated antibodies.

B   Graphs display the FLAG-Orm2 protein levels determined by densitometric quantification of Western blots in cell lysates of WT cells and the indicated mutants 0, 90, and 180 min after the addition of CHX. Each experiment was repeated at least three times. FLAG-Orm2 levels were normalized to Pgk1 loading controls, and each time point was related to $t = 0$ min (set to 1). Data are presented as mean $\pm$ standard deviation.

C   Denaturing immunoprecipitation (IP) of FLAG-Orm2, FLAG-Orm2-3A, and FLAG-Orm2-3D. The control strain expresses untagged Orm2. Input and elutions (with FLAG peptides) were analyzed by SDS–PAGE and Western blot with the indicated antibodies.

D   Epifluorescence and phase-contrast microscopy of living *tul1Δ* mutants expressing GFP-Orm2, GFP-Orm2-3A, or GFP-Orm2-3D (green) with dsRed-HDEL (red). Arrowheads point to GFP-Orm2 accumulations outside the endoplasmic reticulum (ER). Scale bars = 5 µm.

E   SDS–PAGE and Western blot analysis with the indicated antibodies of input and elution (with FLAG peptide) from native anti-FLAG co-immunoprecipitations from cells expressing FLAG-Orm2-K25,33R, FLAG-Orm2-K25,33R-3A, or FLAG-Orm2-K25,33R-3D. The control strain expresses untagged Orm2.

F   The levels of the long-chain bases (LCB) C18-dihydrosphingosine (C18-DHS) and C18-phytosphingosine (C18-PHS) and of two major ceramide (Cer) species in lipid extracts from WT, *tul1Δ*, Orm2-K25,33R, Orm2-3A, and Orm2-3D mutants were measured using LC-MS and quantified using spiked non-yeast LCBs/ceramides as internal standard. Data are normalized to WT levels of the most abundant LCB/ceramide species (set to 1) and presented as mean $\pm$ standard deviation from three independent experiments. Pairwise statistical significance was assessed by Student's *t*-test. *P* values are given above the respective bars.

G   Model for the dual function of the Dsc complex in EGAD-dependent proteasomal as well as ESCRT-dependent lysosomal membrane protein degradation.

# Materials and Methods

### Yeast strains, plasmids, cloning, and growth conditions

*Saccharomyces cerevisiae* strains in this study were SEY6210 derivatives, with exception of the *cim3-1* mutant and its corresponding WT (YP102 background) and the *tor1-1 avo3*$^{\Delta 1274-1430}$ strain (TB50 background). The SGA screen was done in the context of the BY4743 gene deletion collection, using a congenic entry strain (Y7092 background). All yeast strains and genotypes are listed in Appendix Table S6. For liquid cultures, cells were incubated in YNB synthetic medium supplemented with amino acids and 2% glucose at 26°C in a shaker and were grown to midlog phase ($OD_{600}$ = 0.5–0.8) for all experiments. Temperature-sensitive mutants were shifted to restrictive temperature (37°C) as indicated in the figures or figure legends. For growth on agar plates, yeast cells were diluted to $OD_{600\ nm}$ = 0.05 and spotted in serial dilutions on YPD or YNB plates.

Genetic modifications were performed by PCR and/or homologous recombination using standard techniques. Plasmid-expressed genes including their native promoters and terminators were amplified from yeast genomic DNA and cloned into centromeric vectors (pRS series). All N-terminally tagged version of Orm1 and Orm2 (3xHA, 3xFLAG, or GFP) and their respective mutants were typically expressed as the only Orm1 or Orm2 species in cells replacing the endogenous proteins. All constructs were analyzed by DNA sequencing and transformed into yeast cells using standard techniques. Yeast strains and plasmids used in this study as well as primer for PCR-based genetic modifications and cloning are listed in Appendix Table S6.

### Synthetic genetic array screen

Screening for synthetic genetic interactions using the magic marker technology was essentially done as described (Tong *et al*, 2001; Buser *et al*, 2016). In brief, the *VPS4* gene was deleted in the Y7092 strain with a NatR marker to create the entry strain OSY162. OSY162 was mated in two biological replicates to a heterozygous diploid deletion collection (Buser *et al*, 2016) that had been sporulated on

potassium acetate (KOAc, 10 g/l) agar plates for 7 days at 22°C using a Biomek FX 96-well pinning robot (Beckman Coulter). Diploid cells were selected in two consecutive rounds on monosodium glutamate (MSG) minimal medium plates supplemented with geneticin sulfate (G418, 200 mg/l, Werner BioAgents) and nourseothricin (ClonNAT, 100 mg/l, Calbiochem) at 30°C. After recovery on YPD plates (+G418/ClonNAT; 30°C), *vps4Δ* diploids were sporulated on KOAc plates for 7 days at 22°C. Mat.A haploids were first selected on SD minimal medium supplemented with 50 mg/l canavanine (Sigma-Aldrich) and 50 mg/l thialysine (Sigma-Aldrich; 30°C) to suppress A/A diploids, followed by two rounds of haploid selection on MSG plates (+G418/canavanine/thialysine; 30°C) for the presence of the gene deletion introduced by the knockout collection. Finally, double mutants were selected in two consecutive rounds on MSG plates (+G418/ClonNAT/canavanine/thialysine; 30°C). Plates were scanned for documentation and visually inspected for loss or severely impaired growth of strains before and after the double mutant selection. Only hits in both biological replicates were selected. Results are presented in Appendix Table S1.

### Analysis of genetic interaction data

Gene Ontology (GO) enrichment analysis (Harris *et al*, 2004) was performed using 119 genes listed in Appendix Table S1 as synthetically lethal in two biological replicates or lethal in one and sick in the other replicate. They were mapped against the macromolecular complex terms using the Go SLIM mapper of the *Saccharomyces* genome database. GO term fusion was performed based on the Gene Ontology tree (http://www.geneontology.org/). The enriched GO term at the highest level in the GO hierarchy was selected, and its child terms were excluded. In cases when the parent process has a higher *P*-value, the child term was chosen. We also calculate the ratio of the observed (dataset frequency) versus the expected number of genes (genome frequency) associated with the GO term, referred to as enrichment over genome (McClellan *et al*, 2007). Sub-complex terms (e.g., vacuolar proton-transporting V-type ATPase, V1 domain) were omitted in Fig 1B when the full complex (vacuolar proton-transporting V-type ATPase complex) was also significantly scored. The full analysis is presented in Appendix Table S2. Genetic interaction

networks of *tul1Δ* were analyzed from publicly available genetic interaction data using http://thecellmap.org (Usaj *et al*, 2017).

## Evolutionary conservation of Dsc3

*Saccharomyces cerevisiae* Dsc3 (UniProt accession Q12015) is an ubiquitin-like domain protein (residues 25–125) as revealed by a Hidden Markov model (HMM) search against the Pfam domain database version 32 (Finn *et al*, 2016). Its conserved C-terminal region is described in the family-specific domain of unknown function DUF2407_C (residues 153–290) and contains two transmembrane helices (TMHMM; Krogh *et al*, 2001). To collect all family members, NCBI-blastp searches within the UniProt reference proteome database applying highly significant *E*-values ($< 1e^{-5}$) were performed (BLAST 2.6.0+; Camacho *et al*, 2009). Orthologs were identified as reciprocal best blast hit not only in a wide range of fungi, but also in some Alveolata (such as *Oxytricha trifallax*), Amoebozoa (such as *Planoprotostelium fungivorum*), a single entry in plants (*Quercus suber*), and numerous ones in heterokonts (such as *Albugo candida* or *Nannochloropsis gaditana*). They all share the same domain architecture with the ubiquitin-like domain (also classified as DUF2407 in Pfam) and the DUF2407_C domain including the two transmembrane regions at the very C-terminus. To discover similar ubiquitin-like proteins in higher eukaryotes, we performed a NCBI-blastp search with the DUF2407 ubiquitin-like regions applying less restricted *E*-values ($< 0.01$). Among the numerous hits to ubiquitin domains, one protein family was very prominent, and hits were present in a wide range of taxa, not only in mammals but also in sauropsids, frog, fish, and nematodes, the "Transmembrane and ubiquitin-like domain-containing protein 1" (TMUB1) and "Transmembrane and ubiquitin-like domain-containing protein 2" (TMUB2) family. TMUB1/2 proteins are characterized by an amino-terminal transmembrane region, a ubiquitin domain, and two transmembrane regions at the carboxy-terminus. TMUB1 has been shown to be part of the gp78 ERAD ubiquitin ligase complex (Jo *et al*, 2011). TMUB1/2 orthologs were identified not only in many vertebrates, but also in insects (not in *Drosophila*), nematodes, and sea anemones. There is hardly any overlap in the taxonomic distribution of Dsc3 and TMUB1/2 orthologs, besides in some heterokonts, such as *Pythium ultimum*, where a Dsc3 and a TMUB1/2 ortholog were detected. The similarity in domain architecture, a ubiquitin-like domain and two carboxy-terminal transmembrane regions, points to TMUB1/2 proteins as good candidate for adopting Dsc3 function in higher eukaryotes.

## Tul1 protein family and candidates in mammals

*Saccharomyces cerevisiae* Tul1 (UniProt accession P36096) predicted to encode a signal peptide (region 1–26), 7 transmembrane regions (region 399–654), and a carboxy-terminal zf-rbx1 RING-H2 zinc finger domain (region 698–752) (Krogh *et al*, 2001; Petersen *et al*, 2011; Finn *et al*, 2016). Orthologs are widespread among fungi but could as well be detected in Alveolata (such as *Plasmodium falciparum*), Amoebozoa (*Dictyostelium discoideum*), plants (*Arabidopsis thaliana*), and heterokonts (*Thalassiosira pseudonana*) (NCBI-blastp *E*-values < 1e-20). Sequences were selected for a wide taxonomic distribution, aligned using MAFFT (v7.394, (Katoh & Toh, 2008)), and the carboxy-terminal RING finger domain was

extracted with Jalview (Waterhouse *et al*, 2009). For creating the HMM, alignment columns with more than 50% gaps were deleted and the model was searched against the human UniProt reference proteome (21057 entries) with hmmsearch (HMMER 3.2.1, (Eddy, 2011)). The best scoring protein was "RING finger protein 122" (accession Q9H9V4) with an E-value of 5.8e-13 (Fig EV2, Appendix Table S3). The alignment was also used for inferring a maximum-likelihood phylogenetic tree with IQ-tree (version 1.6.7) and for calculating bootstrap values with UFboot2 (Nguyen *et al*, 2015) (Hoang *et al*, 2018).

## Sample preparation for mass spectrometry

For quantitative analysis of protein turnover, isogenic *pep4Δ* and *tul1Δ pep4Δ* mutant yeast cells were grown in synthetic medium (YNB) containing 2% glucose with all required amino acids and "heavy" $[^{13}C_6, ^{15}N_2]$-L-lysine (Sigma) at a final concentration of 20.9 mg/l. Cells were pre-cultured over night in 10 ml medium at 26°C. Afterward, cells were kept at midlog phase for 24 h ($OD_{600} \leq 0.7$) in 50 ml. After three washes at 4°C with cold YNB medium without amino acids, cells were resuspended in 100 ml medium containing light L-lysine to a final concentration of 20 mg/l. After 180 min of incubation in "light" medium, cells (60 $OD_{600}$) were harvested by centrifugation. The cell pellets were resuspended in 500 μl ice-cold 2 M urea, 100 mM $NH_4HCO_3$, pH 8.0, and subjected to five rounds of glass bead lysis at 4°C for 2 min with 2-min recovery on ice between each round. Lysate were cleared twice at 200 × *g* for 5 min at 4°C, and supernatants were stored at −80°C. Protein concentration was determined by Bradford assay (Thermo). For in-solution digestion, 0.6 mg yeast protein extract was reduced with dithiothreitol (10 mM) at room temperature for 60 min. Proteins were digested for 5.5 h at 37°C on an ultrasonic bath by adding 12 μg Lys-C (protease/protein = 1:50). Digested peptides were alkylated with iodoacetamide (50 mM) at room temperature for 20 min. Peptides were separated by reversed-phase chromatography using a Beckman Gold HPLC system (Beckman Coulter, Brea, CA; dual pump model 127S; UV–Vis detector model 168NM). For prefractionation, digested yeast proteins (0.6 mg) were loaded on a XBridge Peptide BEH C18 Column, 300 Å, 5 μm, 4.6 × 250 mm (Waters, Milford, MA, USA), and eluted at a constant flow rate of 0.5 ml/min. Solvents for HPLC were 10 mM ammonium formate, pH 10 in 2% acetonitrile (solvent A), and 98% acetonitrile (solvent B), respectively. Gradient profile was as follows: 0–10 min, 5% B; 10–70 min, 5–35% B; 70–85 min, 35–70% B; 85–95 min, 70% B; and 95–105 min, 100% B. Fraction collection started 1 min after sample injection, and 100 fractions were collected at 1-min intervals. Fractions were lyophilized and stored at −20°C.

## LC-MS/MS analysis

Lyophilized fractions were solved in 18 μl 0.2% formic acid, and 2–3 fractions were pooled for analysis. Samples were analyzed using an UltiMate 3000 RSLCnano-HPLC system coupled to a Q Exactive HF mass spectrometer (both Thermo Scientific, Bremen, Germany) equipped with a Nanospray Flex ionization source. The peptides were separated on a homemade fritless fused-silica microcapillary column (100 × 280 μm × 20 cm) packed with 2.4 μm reversed-phase C18 material. Solvents for HPLC were 0.1% formic acid

(solvent A) and 0.1% formic acid in 85% acetonitrile (solvent B). The gradient profile was as follows: 0–2 min, 4% B; 2–115 min, 4–30% B; 115–120 min, 30–100% B; and 120–125 min, 100% B. The flow rate was set to 300 nl/min. The Q Exactive HF mass spectrometer was operating in the data-dependent mode selecting the top 20 most abundant isotope patterns with charge > 1 from the survey scan with an isolation window of 1.6 mass-to-charge ratio ($m/z$). Survey full-scan MS spectra were acquired from 300 to 1,750 $m/z$ at a resolution of 60,000 with a maximum injection time (IT) of 120 ms, and automatic gain control (AGC) target of 1e6. The selected isotope patterns were fragmented by higher-energy collisional dissociation (HCD) with normalized collision energy of 28 at a resolution of 30,000 with a maximum IT of 120 ms, and AGC target of 5e5.

### Analysis of quantitative proteome data

Data analysis was performed with the Proteome Discoverer 2.2 (Thermo Scientific Version 2.2; RRID:SCR_014477) using the search engine Sequest. The raw files were searched against the orf_trans_all yeast database (6,627 sequences). Precursor and fragment mass tolerance were set to 10 ppm and 0.02 Da, respectively, and up to two missed cleavages were allowed. Carbamidomethylation of cysteine was set as static modification, and oxidation of methionine and heavy lysine ($^{13}C_6\,^{15}N_2$/+8.014 Da) was set as variable modification. Acetylation, methionine loss, and methionine loss plus acetylation were set as N-terminal dynamic modification of proteins. Peptide identifications were filtered at 1% false discovery rate. For stringent analysis, peptides with mixed heavy and light labels were excluded from the quantification and only proteins with a (H/L) ratio < 1 were considered for analysis, which can be expected even for long-lived proteins due to approximate doubling of biomass during the experiment (Christiano *et al*, 2014). When light but no corresponding heavy peptides were detected, the software arbitrarily set the H/L ratio to 0.01. Using these filtering criteria, our analysis quantified 2,511 proteins from two biological replicates (Appendix Table S4). 2,335 proteins had similar H/L ratios (deviation below twofold) in both strains. Proteins with a more then twofold higher H/L ratio in *tul1Δ pep4Δ* compared with *pep4Δ* were considered as stabilized in *tul1Δ pep4Δ* mutants. Transmembrane protein predictions for 76 stabilized proteins were done with Protter v.1.0 (Omasits *et al*, 2014), and Gene Ontology term analysis was done as described above (see section: "Analysis of genetic interaction data").

### Preparation of whole-cell protein extracts

To prepare whole-cell lysates, proteins were extracted by a modified alkaline extraction protocol (Kushnirov, 2000). Cells (4 OD$_{600nm}$) were harvested by centrifugation (5 min, 1,500 × *g*, 4°C) and washed in ice-cold water with phosphatase inhibitors (10 mM NaF, 10 mM β-glycerol phosphate, PhosSTOP; Roche, 1 tablet per 100 ml). After centrifugation (3 min, 15,000 × *g*, 4°C), cells were resuspended in 0.1 M NaOH with the same phosphatase inhibitors and incubated at room temperature for 5 min. After centrifugation (3 min, 15,000 × *g*, 4°C), pellets were resuspended in Läemmli sample buffer, denatured (95°C, 15 min), and cell debris was removed by centrifugation (3 min, 15,000 × *g*, 4°C).

### Western Blot analysis and immunodetection

Protein extracts dissolved in Läemmli sample buffer were separated by SDS–PAGE (Bio-Rad Mini Protean) and transferred to PVDF membranes by semi-dry electroblotting. Antibodies used in this study are listed in Appendix Table S6.

### Immunodetection of ubiquitinated proteins

For detection of ubiquitinated proteins, samples were separated by SDS–PAGE (200 V, 40 mA/gel, 1 h) using 12.5% gels. Proteins were then transferred to equilibrated nitrocellulose membranes (20V, 100 mA/gel, 2 h) at 4°C with MeOH transfer buffer (25 mM Tris–HCl, 192 mM glycine, 20% MeOH). After transfer, membranes were rinsed with deionized water, sandwiched between two wet 3-mm filter papers, submerged with deionized water, and autoclaved at 121°C for 30 min. Afterward, the membrane was blocked with 10% BSA in 0.45% TWEEN-20 for at least 1 h. Membrane has then been incubated with P4D1 anti-ubiquitin antibody overnight in 10% BSA blocking solution.

### Phos-tag SDS–PAGE

Phos-tag gels (50 μM Phos-tag acrylamide (Wako); 100 μm MnCl$_2$) were prepared according to the manufacturer's specifications. Gels were run in a standard Läemmli electrophoresis buffer at 200V, 40 mA for 1.5 h, afterward rinsed in Western blot transfer buffer with 10 mM EDTA for 20 min, and equilibrated in transfer buffer, followed by standard semi-dry electroblotting to PVDF membranes.

### RNA isolation and quantitative PCR (RT–qPCR)

RNA isolation, cDNA synthesis, and quantitative RT–PCR analysis were performed as described previously (Schmidt *et al*, 2017). RNA was extracted from logarithmically growing cells (40 OD) using the Qiagen RNeasy Mini Kit. Yield and purity were determined photometrically. cDNA was prepared from 5 μg DNAse I-treated RNA using the RevertAid First Strand cDNA Synthesis Kit (Thermo Fisher) with oligo-dT primer according to the standard protocol. qPCR was performed in 10 μl scale with 2 μl of cDNA, 5 μl 2xTaqMan qPCR mix (Applied Biosystems), and 0.5 μl TaqMan probe on a PikoReal 96 Real-Time PCR System (Thermo Fisher) with 7 min of initial denaturation (95°C) and 40 cycles of 5 s 95°C and 30 s 60°C. TaqMan gene expression assays were from Thermo Fisher (*ORM2*: Sc04149509_s1; *PGK1*: Sc04104844_s1). RT–qPCR analysis was done from four independent biological samples in three to four technical replicates. Data were analyzed with the PikoReal software (version 2.2; Thermo Scientific) with manual threshold adjustment, and relative mRNA abundance was calculated in Microsoft Excel (Version 16.16.2; RRID:SCR_016137) using the $\Delta\Delta C_T$ method (Livak & Schmittgen, 2001).

### Denaturing FLAG-Orm2 immunoprecipitation

Denaturing immunoprecipitation of FLAG-Orm2 was adapted from Breslow *et al* (2010). Briefly, 20 OD$_{600}$ of logarithmically growing

yeast were collected and resuspended in 1 ml ice-cold water with phosphatase inhibitors (10 mM NaF, 10 mM β-glycerol phosphate, PhosSTOP; Roche, 1 tablet per 100 ml). Cell pellets were resuspended in 1 ml 10% trichloroacetic acid (TCA) and incubated for at least 30 min at 4°C or stored at −20°C. Cell pellets were washed twice in 1 ml acetone and then dried for 2 min without heating in a speed vac. 100 μl SDS lysis buffer (50 mM Tris–HCl pH 8, 1 mM EDTA, 1% SDS, 2 M urea, 10 mM NaF) supplemented with protease inhibitors (1× cOmplete® EDTA-free (Roche), 4 mM PMSF, 20 mM N-ethylmaleimid, 1× yeast protease inhibitor mix (Sigma-Aldrich)) was added, and the dried pellets were solubilized by vortexing with glass beads (0.75–1 mm; 4 × 2 min at 4°C). Lysates were diluted by addition of 900 μl immunoprecipitation (IP) buffer (100 mM Tris–HCl pH 8, 100 mM NaCl, 1.5% Triton X-100, supplemented with protease inhibitors) and cleared by centrifugation (15 min; 20,000 $g$; 4°C). Supernatants were then added to anti-Flag magnetic beads (Sigma-Aldrich, Cat. # M8823; RRID:AB_2637089) that were prewashed in IP buffer, and immunoprecipitated for 2.5–3 h at 4°C. The beads were washed twice for 15 min at 4°C with Triton wash buffer (100 mM Tris–HCl pH 8, 200 mM NaCl, 1.5% Triton X-100) and twice with TWEEN wash buffer (50 mM Tris–HCl pH 7.5, 150 mM NaCl; 0.1% TWEEN-20) for 15 min at room temperature. Proteins were eluted by addition of TWEEN wash buffer supplemented with 500 ng/ml 3xFLAG peptide (Sigma-Aldrich) and incubated for 30 min at room temperature. Magnetic beads were removed, and the eluate was supplemented with 5× sample buffer (250 mM Tris–HCl, pH 6.8, 10% SDS, 50% glycerol, 10% β-mercaptoethanol) and denatured for 10 min at 95°C.

### Non-denaturing FLAG-Orm2 immunoprecipitation

Non-denaturing isolation of FLAG-Orm2 was performed essentially as described (Breslow *et al*, 2010) from cryo-ground whole cells instead of microsomes. Logarithmically growing cells (1,000 OD$_{600 \, nm}$) were harvested by centrifugation, and pellets were frozen in liquid nitrogen. Frozen cells were ground in a 6770 freezer mill (SPEX SamplePrep) with liquid nitrogen cooling. 1 g yeast powder was resuspended in 5 ml lysis buffer (50 mM HEPES/KOH, pH 6.8; 150 mM KOAc, 2 mM Mg(OAc)$_2$; 1 mM CaCl$_2$; 15% glycerol) supplemented with 2% digitonin (Sigma-Aldrich); protease and phosphatase inhibitor (10 μg/ml leupeptin; 1 μg/ml aprotinin; 1 μg/ml pepstatin A; 0.4 mM PEFA-Bloc; 1× yeast protease inhibitor mix (Sigma-Aldrich); 1x cOmplete EDTA-free (Roche); 1× PhosSTOP (Roche)), and incubated rotating at 4°C for 1.5 h. Unsolubilized material was removed by centrifugation (2,000 × $g$; 15 min; 4°C), and the lysate was cleared by ultracentrifugation (44,000 × $g$; 30 min; 4°C; TLA55 rotor). The supernatant (4 ml) was mixed with an equal volume of lysis buffer with 0.1% digitonin, added to equilibrated anti-FLAG magnetic beads (Sigma-Aldrich Cat. # M8823; RRID:AB_2637089), and rotated for 3 h at 4°C. Beads were recovered with a magnetic rack and washed eight times for 10 min with 5–10 ml lysis buffer containing 0.1% digitonin. Proteins were eluted twice with 150 μl 50 mM HEPES/KOH, pH 6.8; 150 mM NaCl, 15% glycerol, 0.1% digitonin, 1 mg/ml 3xFLAG peptide (Sigma-Aldrich) at 4°C for 15 min, and elutions were pooled, supplemented with SDS–PAGE sample buffer, and denatured (95°C, 10 min).

### Live-cell fluorescence, wide-field and confocal microscopy

For microscopy, cells were grown to midlog (OD$_{600}$ = 0.5–0.8) phase in YNB media, concentrated by centrifugation and directly mounted onto glass slides. For confocal microscopy, glass slides were coated with concanavalin A (Sigma, Cat.#L7647). Live-cell wide-field fluorescence microscopy was carried out using a Zeiss Axio Imager M1 equipped with a sola light engine LED light source (Lumencore), a 100× oil immersion objective (NA 1.45) standard GFP and mCherry fluorescent filters, a SPOT Xplorer CCD camera, and Visitron VisiView software (version 2.1.4). Single-plane confocal images and Z stacks were acquired at RT using a Laser Scanning Confocal Microscope (SP8, Leica) equipped with a gated white light laser (excitation at 488 nm (GFP) and 558 (dsRed)), a 63× glycerol immersion objective (NA 1.3), and the LasX software (v.3.5.2.18963). Images were recorded in resonance scanning mode (8,000 Hz) using hybrid detectors at 256 × 256 pixels (pixel size x, y 60,304 nm; z 332,833 nm; line averaging 10). The confocal images were deconvolved with Huygens Professional version 18.10 (Scientific Volume Imaging, the Netherlands, http://svi.nl), using the CMLE algorithm, with SNR:10 and 40 iterations at otherwise default settings. Single z planes were extracted in tiff format with Fiji software (v 1.0) (Schindelin *et al*, 2012). The brightness and contrast of the images (from wide-field and confocal microscopy) in the figures were adjusted using Photoshop CS5 (Adobe Version 12.0.4x64; RRID:SCR_014199). For merged images, the levels of red and green channels were separately adjusted.

### Cycloheximide chase assay

Logarithmically growing cells (20 OD$_{600 \, nm}$) were harvested by centrifugation. *cdc48-3* and *CDC48*-WT cells were pre-shifted to 37°C for 2 h, and *sec13-4* and *SEC13*-WT cells for 30 min. Cells were resuspended in 50 ml fresh medium (for experiments with temperature-sensitive mutants, the medium was pre-warmed at 37°). For chemical proteasome inhibition, *pdr5Δ* cells were pre-incubated with MG-132 (50 μM) for 10 min. For the inhibition of TORC2 signaling in the *tor1-1 avo3*$^{Δ1274-1430}$ strain, rapamycin (200 nM; stock 1 mM in methanol; LC laboratories) or the same volume of vehicle were added at $t$ = 0 min. At $t$ = 0 min, 10 ml (4 OD$_{600 \, nm}$) was harvested by centrifugation, washed once with ice-cold water containing phosphatase inhibitors (see above), and pellets were snap-frozen in liquid nitrogen. To the remaining culture, 50 μg/ml cycloheximide (Sigma-Aldrich) was added from a 10 mg/ml stock. After the indicated time points, 10 ml culture was harvested, washed, and frozen as above. Whole-cell extracts were prepared by alkaline extraction. SDS–PAGE, Western blot detection, and quantification were done as described above.

### Subcellular fractionation and carbonate extraction

The subcellular fractionation protocol was adapted from Neal *et al* (2018). Briefly, 20 OD$_{600 \, nm}$ (40 OD$_{600 \, nm}$ of MG-132-untreated cells in Fig 5C) of logarithmically growing yeast were collected and resuspended in 1 ml ice-cold water with phosphatase inhibitors (10 mM NaF, 10 mM β-glycerophosphate, 0.1× PhosSTOP (Roche)). Cell pellets were resuspended in 500 μl XL buffer (0.24M sorbitol, 1 mM EDTA, 20 mM KH$_2$PO$_4$/NaOH, pH 7.5), supplemented with

100 µl 0.75–1 mm glass beads, and vortexed (4 × 1 min at 4°C). Cell lysates were cleared at 2,500 g for 5 min. The supernatant was ultracentrifuged at 100,000 g for 1–1.5 h to separate pellet (P100) and supernatant (S100) fractions. The S100 fraction was removed, and 10% (final) trichloroacetic acid (TCA) was added. The P100 was washed once in cold XL buffer without disturbing the pellet and centrifuged at 13,000 g for 5 min. The supernatant was discarded. The washed P100 fraction was resuspended in 100 µl SDS lysis buffer (50 mM Tris–HCl pH 8, 1 mM EDTA, 1% SDS, 2 M urea, 10 mM NaF), supplemented with protease inhibitors (1× EDTA-free protease inhibitor tablet, 1 mM PMSF, 10 mM NEM, yeast protease inhibitor cocktail), and 1 ml of 10% ice-cold TCA was added. For carbonate extraction (Fujiki *et al*, 1982), the washed P100 fraction was resuspended in 200 mM $Na_2CO_3$ solution (adjusted to the indicated pH with 5N NaOH) and incubated on ice for 20 min. Subsequently, the extraction was again ultracentrifuged at 100,000 g for 45 min to separate pellet (P100) and supernatant (S100) fractions. S100 fractions were precipitated with 20% TCA. P100 fractions were resuspended in 100 µl SDS lysis buffer supplemented with protease inhibitors and subsequently precipitated with 20% TCA. All TCA protein extracts were solubilized in 100 µl SDS lysis buffer supplemented with protease inhibitors. Immunoprecipitations under denaturing conditions were described above.

## Lipid extraction and mass spectrometry

The analysis of ceramides (Fig 6A) was performed as described previously (Frohlich *et al*, 2015). Briefly, ceramides were analyzed by LC-MS/MS on a Q Exactive *Plus* Orbitrap instrument (Thermo) directly connected to an Agilent 1100 HPLC. Ceramides were extracted from yeast cells according to 75 µg of protein by dichloromethane/methanol extraction (Sullards *et al*, 2011). Prior to extraction, a standard mix containing two ceramides (CER 18:0/17:1) was spiked into each sample for normalization and quantification. 5 µl of each sample was injected onto an Accucore (Thermo Scientific) C18 LC Column (2.1 × 150 mm × 2.6 µm particle size). Lipids were fractionated by reverse-phase chromatography over a 30-min linear gradient (mobile phase A: acetonitrile/water (50:50 v/v), 10 mM ammonium formate, 0.2% formic acid; mobile phase B: methanol/isopropanol/water (10:88:2 v/v/v), 2 mM ammonium formate, 0.01% formic acid) from 40% B to 100% B.

For combined LCB and ceramide analysis (Fig 7F), lipids were extracted from lysed yeast cells according to 150 µg of protein by chloroform/methanol extraction (Ejsing *et al*, 2009). Prior to extraction, a standard mix containing sphingosine (LCB 17:0) and ceramide (CER 18:0/17:1) was spiked into each sample for normalization and quantification. Dried lipid samples were dissolved in a 65:35 mixture of mobile phase A (60:40 water/acetonitrile, including 10 mM ammonium formate and 0.1% formic acid) and mobile phase B (88:10:2 2-propanol/acetonitrile/$H_2O$, including 2 mM ammonium formate and 0.02% formic acid). HPLC analysis was performed employing a C30 reverse-phase column (Thermo Acclaim C30, 2.1 × 250 mm, 3 µm, operated at 50°C; Thermo Fisher Scientific) connected to an HP 1100 series HPLC (Agilent) HPLC system and a Q Exactive *Plus* Orbitrap mass spectrometer (Thermo Fisher Scientific) equipped with a heated electrospray ionization (HESI) probe. The elution was performed with a gradient of 45 min; during 0–3 min, elution starts with 40% B and increases to 100%, in a linear gradient

over 23 min. 100% B is maintained for 3 min. Afterward, solvent B was decreased to 40% and maintained for another 15 min for column re-equilibration. The mass spectrometer was run in negative-ion mode. The scan rate was set from 200 to 2,000 *m/z*. Mass resolution was 70,000 with an AGC target of 3,000,000 and a maximum injection time of 100 ms. The MS was operated in data-dependent mode. For LC-MS/MS, the resolution was 17,500 with a maximum injection time of 50 ms and an AGC target of 10,000. The loop count was 10. Selected ions were fragmented by HCD (higher-energy collision dissociation) with a normalized collision energy of 30. The dynamic exclusion list was set to 10s to avoid repetitive sequencing. Ceramide peaks were identified using the Lipid Search algorithm (MKI, Tokyo, Japan). Peaks were defined through raw files, product ion, and precursor ion accurate masses. Ceramides were identified by database (> 1,000,000 entries) search of negative-ion adducts. The accurate mass extracted ion chromatograms were integrated for each identified lipid precursor and peak areas obtained for quantitation. Internal standards were used for normalization and to calculate absolute values (in pmol/µg protein). To compare data from different experimental setups, data were normalized to WT levels of the most abundant LCB and ceramide species (C18-PHS (phytosphingosine) and Cer44:0;4). Data are presented as mean ± standard deviation from three independent experiments.

## *In vivo* sphingolipid labeling

Sphingolipid labeling and extraction were done as described previously (Tabuchi *et al*, 2006). Logarithmically growing cells (5 $OD_{600nm}$) were harvested by centrifugation and resuspended in 470 µl fresh medium. 30 µCi of [³H]-L-serine (1 µCi/µl; Hartmann Analytic) was added, and cells were incubated in a thermomixer at 30°C, 700 rpm for 5 h. Proteins were precipitated with 4.5% perchloric acid, and pellets were washed with ice-cold 100 mM EDTA, resuspended in 50 µl water, and subjected to mild alkaline methanolysis (50% methanol; 10% 1-butanol; 10% monomethylamine; 50 min at 50°C) of ester lipids. Lysates were vacuum-dried in a speed vac (37°C), pellets were resuspended thoroughly in 300 µl water by sonication, and total label incorporation was assessed in duplicates by szintillation counting for normalization (LS6500, Beckmann Coulter). Sphingolipids were extracted three times with water-saturated 1-butanol, vacuum-dried in a speed vac (37°C), and dissolved in 50 µl chloroform: methanol: water 10:10:3 for thin-layer chromatography.

## Thin-layer chromatography

Thin-layer chromatography (solvent system chloroform: methanol: 4.2N ammonium hydroxide 9:7:2) was done as described (Tabuchi *et al*, 2006) on aluminum silica TLC plates (Sigma) for 75 min. Subsequently, plates were dried under air flow and treated twice with En³Hance autoradiography enhancer (Perkin-Elmer), dried again, and exposed to autoradiography films (CL-XPosure Film, Thermo) at −80°C. The identity of labeled bands was confirmed with specific mutant yeast strains or inhibitors (data not shown). A tritium-labeled band X close to the solvent front that was observed previously (Tabuchi *et al*, 2006) is not sensitive to myriocin treatment (Sigma-Aldrich, 1.5 µM), suggesting that it is not a sphingolipid.

### Quantification and statistical analysis

Statistical details and sample numbers of quantitative analyses can be found in the respective figures and corresponding figure legends. Quantitative data are usually displayed as mean ± standard deviation from at least three biological replicates and relate to the wild-type control.

#### GO analysis of genetic interaction data

A hypergeometric test was used to estimate whether the mapped GO term is significantly enriched with the selected genes. The null hypothesis is that the selected genes are randomly sampled from all yeast genes. The resulting *P*-values were corrected with the Benjamini–Hochberg method. All adjusted *P*-values below 0.05 were reported.

#### GO analysis of proteome data

A hypergeometric test was used to estimate whether the mapped GO term is significantly enriched with the selected genes. The null hypothesis is that the selected genes are randomly sampled from all yeast genes. The resulting *P*-values were corrected with the Benjamini–Hochberg method. All adjusted *P*-values (false discovery rate) below 0.1 were reported.

#### Quantification of Western Blot analysis

Western blot signals were quantified by densitometry using ImageJ2 (Version 2.0.0-rc49/1.51 h; RRID:SCR_003070) (Rueden *et al*, 2017), quantifications were exported to Microsoft Excel (Version 16.16.2; RRID:SCR_016137), normalized to the respective Pgk1 loading controls, and presented as mean ± standard deviation from at least three independent experiments. $t = 0$ was set to 1.

#### Quantitative PCR (RT–qPCR)

Mean, normalized $\Delta\Delta C_T$ values were individually calculated for four independent biological replicates (each with three to four technical replicates), log2-transformed to calculate fold change, and presented as mean fold change over WT ± standard deviation. To assess statistical significance, a one-sided Student's *t*-test was used on the mean $\Delta\Delta C_T$ values (i.e., prior to log transformation) of each of the four biological replicates.

#### Lipid extraction and mass spectrometry

Internal standards for ceramides (Cer 18:1;2/17:0;0) and LBCs (LCB 17:0) spiked in prior to extraction were used for normalization and to calculate absolute values (in pmol/μg protein). To compare data from different experimental setups, data were normalized to WT levels of the most abundant LCB and ceramide species (C18-phytosphingosine and Cer44:0;4). Data are presented as mean ± standard deviation from three independent experiments. Student's *t*-test was used on the absolute values to assess statistical significance.

## Data and software availability

All relevant data have been included in the paper in main and expanded view figures and Supporting Information.

**Expanded View** for this article is available online.

## Acknowledgements

We would like to thank Theresa Dunn, Peter Espenshade, Kai-Uwe Fröhlich, Ben Glick, Robbie Loewith, Ming Li, Carl Mann, Howard Riezman, Chris Stephan, and Jonathan Weissman for providing reagents. Work in the Teis laboratory was supported by the Austrian Science Fund (FWF-Y444-B12, P30263, P29583) and MCBO (W1101-B18) to D.T and EMBO/Marie Curie (ALTF 642-2012; EMBOCOFUND2010, GA-2010-267146), MUI-START (2013042023), and "Tiroler Wissenschaftsfond" to O.S. Work in the Fröhlich laboratory is supported by a DFG grant FR 3647/2-1 and the SFB 944. Work in the Peter laboratory was supported by funding from the European Research Council (ERC), the Swiss National Science Foundation (SNSF), and ETH Zürich.

## Author contributions

OS performed the SGA analysis with MP and the SL-TLC analysis; OS, YW, VB, and MAW constructed yeast strains and plasmids and conducted microscopy, biochemical, and genetic experiments; MA performed the statistical analysis of the GO dataset; SE and FF analyzed ceramide and LCB levels; AS analyzed evolutionary conservation of Dsc subunits; LK and HL did the mass spectrometry analysis; OS and DT analyzed data; DT conceptualized and guided the study; and DT and OS wrote the manuscript.

## Conflict of interest

The authors declare that they have no conflict of interest.

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
