## [Review Process File · The EMBO Journal]

Endosome and Golgi-associated degradation (EGAD) of membrane proteins regulates sphingolipid metabolism

Oliver Schmidt, Yannick Weyer, Verena Baumann, Michael A. Widerin, Sebastian Eising, Mihaela Angelova, Alexander Schleiffer, Leopold Kremser, Herbert Lindner, Matthias Peter, Florian Fröhlich, and David Teis.

Review timeline:

Submission date:	22 nd December 2018
Editorial Decision:	5 th February 2019
Revision received:	18 th April 2019
Editorial Decision:	7 th May 2019
Revision received:	8 th May 2019
Accepted:	8 th May 2019

Editor: Hartmut Vodermaier

Transaction Report:

1st Editorial Decision

5th February 2019

Thank you again for submitting your manuscript on post-ER organelle-associated membrane protein degradation for our editorial consideration. I have now received the below-copied reports from three expert referees, who all consider your findings significant and mostly well-supported by the data. In light of these positive assessments, we shall therefore be happy to publish the study in The EMBO Journal after adequate revision of a number of specific technical points and presentational issues.

REFeree REPORTS

Referee #1:

The manuscript from Schmidt and colleagues represents a major step forward in our understanding of regulated protein degradation and proteostasis. The authors define a new pathway for membrane protein degradation via the proteasome (pERAD). They identify the lipid homeostasis regulator Orm2 as the first post-ER substrate for the Golgi/endosomal Dsc E3 ligase in budding yeast. While the E3 ligase complex has been shown to act on plasma membrane substrates this is the first example of an ER protein. The authors beautifully dissect the mechanism by which Orm2's degradation is regulated, showing that Orm2 TORC2-dependent phosphorylation and ubiquitylation are required. The physiological importance of the pERAD pathway is highlighted by the demonstration that lipid homeostasis is disrupted in pERAD mutants. Importantly, proteomic and genetic studies indicate that many more pERAD substrates exist and homology searches point to the presence of a parallel pathway in mammalian cells. The data fully support the conclusions of what will likely be a landmark paper in the field.

Major comments:

This is an outstanding study, beautifully executed.

Minor comments/suggestions:

1. Consider adding "in yeast" to the title. It adds important information for the casual reader and does not reduce the impact of the study on eukaryotic biology.
2. Line 264, should read Figure 2D
3. Line 311, phrase "not the only one" is unclear. Please explain why it would not be the only one more clearly or remove.

Referee #2:

Schmidt et al. describe a previously uncharacterized protein degradation pathway in post-ER membrane compartments, termed pERAD. Using genetic and proteomics screens, the authors show that Orm2, a negative regulator of sphingolipid biosynthesis, is turned over by the Dsc ubiquitin ligase complex. In contrast to previously described Dsc complex substrates, poly-ubiquitinated Orm2 is degraded by the proteasome and its membrane extraction requires the Cdc48 ATPase complex. In this context, the Dsc complex appears to function analogously to ERAD complexes but in post-ER compartments. In the case of Orm2, Dsc-mediated degradation requires Orm2 export from the ER which appears to be stimulated by phosphorylation. Finally, it is shown that deletion of Dsc components results in Orm2 accumulation and decreased ceramide synthesis.

Overall, the data presented by Schmidt et al. is of good quality and the underlying logic of the work is well articulated. The identification of Dsc complex in proteasome-dependent degradation of membrane proteins and the characterization of Orm2 as one of its substrates is certainly of interest for the broad readership of EMBO Journal and I would enthusiastically support its publication.

However, there are a few points that if clarified could improve the manuscript.

The lower amounts of ceramides in *Tul1Δ* mutants (Fig 6A) is interpreted as a consequence of increased Orm2 levels. But how much higher are Orm2 levels in the ER of cells? According to Figures 3A or 7D, most Orm2 appears in post ER compartments in *Tul1Δ* cells, from where it is not plausible that it inhibits SPT and sphingolipid synthesis. Along these lines, it is reasonable to assume that Orm2 ER export is sufficient to de-repress SPT activity. Then, why the need of *Tul1*-mediated degradation? Do the two mechanisms (ER export and *Tul1*-mediated degradation) serve different purposes? Is it possible that Orm2 can be retrieved back to the ER? In my opinion these are relevant issues that should be properly discussed in the manuscript.

Still related with the sphingolipid levels in the *Tul1* mutant, is there any information on how these are affected by the *Orm2-3A* allele, which appears to localize predominantly to the ER and less sensitive to *Tul1* degradation? Similarly, does this allele have a synthetic growth defect with *Vps4Δ*? The effect of *Orm2Δ* on *Tul1ΔVps4Δ* growth defect (Fig 3C) is negligible, suggesting that other *Tul1* substrates are more relevant to the growth phenotype. In face of the data presented, I would recommend to tone down the conclusions on the effect of Orm2 on the growth defect *Tul1ΔVps4Δ*.

Minor points:

- Line 295-297, *Gld1* and *Vld1* trafficking pathways are vaguely introduced. One-line explaining would be helpful for non-experts.
- Figure 3C, in *Tul1ΔVps4Δ* double mutants, GFP-Orm2 accumulated in class E compartments, but why is Orm2 depleted from the ER in these mutants? Do the levels of Orm2 differ from *Vps4Δ* cells and to *Tul1ΔVps4Δ*? This should be clarified.
- Figure 6E, the levels of *Orm2-3A* protein appear slightly higher than Orm2 (quantification would be appropriate), as it would be expected from the model proposed. However the levels still respond to the presence of *Tul1*. How can this be explained? It would be important to point this out in the text and speculate about possible reasons.
- The references throughout the manuscript are erratic, sometimes absent (e.g. line 84), other times overly used (e.g. line 191-194) and in other instances misused (e.g. line 335 Baldrige & Rapoport 2016- Bodnar & Rapoport 2017 would be more appropriate).
- The term pERAD is confusing, as it suggests degradation occurring at the ER. Another nomenclature would be advised.
- Proteostasis is sometimes broadly used, e.g. line 154.
- The mass spectrometry results only show data on ceramides. It would be interesting if it also included long chain bases and other sphingolipids.

- Differences were found in Orm2 phosphorylation mediated by Ypk1 versus by Npr1 (Figure 7A and B, line 420-424). These results are not interpreted or discussed.
- The discussion overly focuses on homology analysis of Dsc complex proteins. Perhaps this could be substantially reduced and include the above mentioned points.

Referee #3:

Schmidt et al. present data for a membrane protein degradation pathway that operates primarily in an organelle(s) after the ER (post-ERAD). The pathway was found through a genetic screen for negative synthetic interactions with vps4, a mutant in the ESCRT pathway. A key hit in the screen was the Tul1 RING ubiquitin ligase and an associated subunit from the DSC ubiquitination complex. This had previously been shown to ubiquitinate substrates in the Golgi and endosome and cause their degradation via the ESCRT pathway. However, Schmidt et al. find that the model transmembrane substrate, Orm2, that they identified through proteomic analysis was extracted from membranes by Cdc48 and degraded by the 26S proteasome. Disruption of Orm2 ubiquitination and degradation interfered with sphingolipid homeostasis.

This is a very thorough analysis, and although hints of this post-ERAD pathway existed before (Hwang et al. EMBO J. 2016 found that fission yeast SREBP can be degraded through DSC and the proteasome when a cofactor is deleted), the current study makes a much fuller case. Overall, I think this study is suitable for publication in the EMBO Journal with just some modest changes.

Comments/questions:

Localization data are not quantified. The authors should at least state explicitly that the localization phenotypes they see are fully penetrant. Also, it is known that DSC ligase inhibition in *S. pombe* blocks ER-to-Golgi transport of the DSC complex (see Hwang et al). I was curious if the same occurs in *S. cerevisiae* (not essential to test here but might get at the question of why ERAD E3s apparently cannot compensate for DSC loss).

In Fig. 6A, are these differences all statistically significant? I don't see any statistical analysis here, and some of the differences are modest.

Line 404, Figure 6C-D: Is the fraction of Orm2 in the phosphorylated state increased relative to unmodified protein?

To more convincingly rule out a role for known ERAD pathways in Orm2 degradation, it would be worth testing a hrd1 doa10 double (or possibly hrd1 doa10 asi1 triple) mutant.

The use of commas between allele names when referring to multiply mutant yeast strains (not what most yeasties would recommend) makes the text hard to follow in places, such as lines 308-9.

The references are in general very thorough, but for one ERAD E3 ligase, MARCH6/TEB4 (Doa10 in yeast), there are no/almost no references.

Point-by-point response:**Referee #1:**

The manuscript from Schmidt and colleagues represents a major step forward in our understanding of regulated protein degradation and proteostasis. The authors define a new pathway for membrane protein degradation via the proteasome (pERAD). They identify the lipid homeostasis regulator Orm2 as the first post-ER substrate for the Golgi/endosomal Dsc E3 ligase in budding yeast. While the E3 ligase complex has been shown to act on plasma membrane substrates this is the first example of an ER protein. The authors beautifully dissect the mechanism by which Orm2's degradation is regulated, showing that Orm2 TORC2-dependent phosphorylation and ubiquitinylation are required. The physiological importance of the pERAD pathway is highlighted by the demonstration that lipid homeostasis is disrupted in pERAD mutants. Importantly, proteomic and genetic studies indicate that many more pERAD substrates exist and homology searches point to the presence of a parallel pathway in mammalian cells. The data fully support the conclusions of what will likely be a landmark paper in the field.

Major Comments: This is an outstanding study, beautifully executed.

Response: We would like to thank the reviewer for the positive comments.

Minor comments/suggestions:

Comment: Consider adding "in yeast" to the title. It adds important information for the casual reader and does not reduce the impact of the study on eukaryotic biology.

Response: We would not mind adding 'in yeast' to the title but EMBO J policy limits titles to 100 characters. Based on a request from reviewer 2 we had to change the title of our manuscript. Reviewer 2 recommended to avoid the name pERAD, as it may suggest ER based processes. We agree that the name could be confusing and therefore suggest another (hopefully better) name: endosome and Golgi associated degradation – EGAD.

Comment: Line 264, should read Figure 2D

Response: thank you –we have corrected the mistake

Comment: Line 311, phrase "not the only one" is unclear. Please explain why it would not be the only one more clearly or remove.

Response: We have rephrased in the manuscript the respective paragraph:

The deletion of *ORM2* hardly restored the growth of *tul1Δ vps4Δ* double mutants (Figure 3C). The *vps4Δ tul1Δ orm2Δ* triple mutants still grew poorly when compared to *vps4Δ orm2Δ* and *tul1Δ orm2Δ* double mutants (Figure 3C). It seemed that, in addition to Orm2 other critical substrates of the Dsc complex were degraded by ESCRT-independent pathways.

Referee #2:

Schmidt et al. describe a previously uncharacterized protein degradation pathway in post-ER membrane compartments, termed pERAD. Using genetic and proteomics screens, the authors show that Orm2, a negative regulator of sphingolipid biosynthesis, is turned over by the Dsc ubiquitin ligase complex. In contrast to previously described Dsc complex substrates, poly-ubiquitinated Orm2 is degraded by the proteasome and its membrane extraction requires the Cdc48 ATPase complex. In this context, the Dsc complex appears to function analogously to ERAD complexes but in post-ER compartments. In the case of Orm2, Dsc-mediated degradation requires Orm2 export from the ER, which appears to be stimulated by phosphorylation. Finally, it is shown that deletion of Dsc components results in Orm2 accumulation and decreased ceramide synthesis.

Overall, the data presented by Schmidt et al. is of good quality and the underlying logic of the work is well articulated. The identification of Dsc complex in proteasome-dependent degradation of membrane proteins and the characterization of Orm2 as one of its substrates is certainly of interest for the broad readership of EMBO Journal and I would enthusiastically support its publication. However, there are a few points that if clarified could improve the manuscript.

Response: We would like to thank the reviewer for the positive comments and the insightful suggestions.

Comment: The lower amounts of ceramides in *tul1Δ* mutants (Fig 6A) is interpreted as a consequence of increased Orm2 levels. But how much higher are Orm2 levels in the ER of cells? According to Figures 3A or 7D, most Orm2 appears in post ER compartments in *tul1Δ* cells, from where it is not plausible that it inhibits SPT and sphingolipid synthesis. Along these lines, it is reasonable to assume that Orm2 ER export is sufficient to de-repress SPT activity. Then, why the need of Tul1-mediated degradation? Do the two mechanisms (ER export and Tul1-mediated degradation) serve different purposes? Is it possible that Orm2 can be retrieved back to the ER? In my opinion these are relevant issues that should be properly discussed in the manuscript.

Response: To address these questions we performed additional experiments:

(i) Fluorescence live cell (improved wide field microscopy and new confocal microscopy)

clarifies that failure to degrade Orm2 results in its accumulation at the ER and on post-ER compartments (Figure 3A, EV3A and Figure 4E).

(ii) Biochemical experiments demonstrate that accumulating Orm2 interacts with the ER resident catalytic subunits of the serine palmitoyl transferase (SPT) (Figure 6C and Figure 7E).

(iii) Orm2 accumulation therefore represses sphingolipid synthesis as shown by the additional analysis of LCB and ceramide species (Fig. 6A, B and Figure 7F).

We advance our case through the following experiments:

(1) In the images provided in the first version of the manuscript, it was indeed difficult to evaluate the ER localization of Orm2 in the wide-field fluorescence microscopy. We focused on the accumulation of GFP-Orm2 on endosomes and Golgi, which blurred the signal of GFP-Orm2 at the ER. However, in almost every cell GFP-Orm2 was detected at the ER and on post-ER compartments. We now provide improved live cell wide-field fluorescence images (including quantification) (Figure 3A, 4E) as well as live cell confocal microscopy experiments (Figure EV3A). In single z-sections of WT cells, GFP-Orm2 is exclusively detected at the nuclear and the peripheral ER together with dsRed-HDEL (Figure EV3A). In *tull1Δ* GFP-Orm2 is detected at the ER and on post-ER compartments (Figure 3A, 4E, Figure EV3A).

(2) The presence of ‘non-degradable’ Orm2 at the ER and on post-ER compartments suggested that it could be retrieved back to the ER. Provided that non-degradable Orm2 (e.g. in *tull1Δ* mutants) continuously shuttles between the ER and post-ER compartments, the acute inhibition of COPII transport (using a *sec13-4* temperature sensitive mutant) should trap it at the ER and therefore deplete it from post-ER compartments. Indeed, in *sec13-4* mutants at the non-permissive temperature Orm2 was no longer exported from the ER and not degraded (Figure 4C and Figure EV4E,F). Moreover, in *sec13-4 tull1Δ* mutants GFP-Orm2 was completely depleted from post-ER compartments and only detected at the ER at the non-permissive temperature (Figure 4E). These results show that the failure to degrade Orm2 can result in the retrieval of Orm2 back to the ER. During the same time, the localization of Vps4-GFP to endosomes (which co-localized partially with Orm2 in *tull1Δ* mutants (see Figure 3B)) was unchanged (Figure EV4F).

(3) Accumulating Orm2 (Orm2-K25,33R) co-immunoprecipitated efficiently the ER-resident catalytic subunits Lcb1/2 of SPT (Figure 6C). Since the interaction of Orm2 with SPT is known to inhibit SPT activity (Breslow et al 2010, Nature, PMID: 20182505), this result explains how the accumulation of non-degradable Orm2 (e.g.: Orm2-K25,33R) represses the synthesis of LCB and ceramide species at the ER.

(4) Based on these results, we hypothesized that Orm2-3D, which is constitutively exported from the ER and degraded (Figure 6E and Figure 7D), should no longer interact efficiently with Lcb1/2. In contrast, Orm2-3A, which is trapped at the ER and only poorly degraded (Figure 6E and Figure 7A-D) should interact with Lcb1/2, similar to Orm-K25,33R, which localizes to the ER and to post-ER compartments (Figure 3A).

To directly test this prediction, we first needed to prevent the constitutive degradation of Orm2-3D. Therefore, we introduced K25,33R mutations in Orm2-3D and Orm2-3A. In this setting, the protein levels of Orm2-K25,33R-3D increased to levels that are comparable to Orm2-K25,33R and Orm2-K25,33R-3A (Figure 7E). Orm2-K25,33R-3D only poorly co-immunoprecipitated Lcb1/2 (Figure 7E), because it was continuously exported from the ER (Figure EV5E). Yet, it interacted efficiently with the Dsc complex subunits on Golgi and endosomes (Figure 7E). Conversely, Orm2-K25,33R-3A, which was only detected in the ER (Figure EV5E), co-immunoprecipitated efficiently Lcb1/2, but it only poorly interacted with the subunits of the Dsc complex (Figure 7E). Orm2-K25,33R co-immunoprecipitated Lcb1/2 in amounts comparable to Orm2-K25,33R-3A and the subunits of the Dsc complex (Figure 7E).

These results confirm our key prediction and show that the failure to export Orm2 from the ER and the failure to degrade Orm2 on post ER compartments both increase the interaction with SPT at the ER and therefore should repress sphingolipid biosynthesis to similar extents.

(4) Indeed the synthesis of LCB and ceramide species was repressed in *tull1Δ* mutants and in cells expressing Orm2-3A and Orm2-K25,33R (Figure 6A and Figure 7F). To compare the two independent lipid analysis experiments presented in Figure 6A and Figure 7F, we normalized the values for ceramides to Cer44:0;4 and for LCBs to C18-PHS in WT cells. This was necessary due to two different lipid analysis protocols (one of which better suited to also extract LCBs), new pre-separation columns/gradients and difference in the batches for the

ceramide standards and their detection. Regardless of the different absolute values, the relative differences between WT cells and the mutants were comparable (Figure 6A and Figure 7F). In each experiment, *tul1Δ* mutants and cells expressing Orm2-3A or Orm2-K25,33R had lower levels of LCB and ceramide species than WT cells (Figure 6A and Figure 7F). These results are comparable to the defects in LCB and ceramide synthesis caused by the strong over-expression of Orm2 or by an Orm1/2 double phospho-mutant strain (Breslow et al., 2010, Nature, Fig. 1b and Fig. 5d in their paper). In contrast, Orm2-3D de-repressed sphingolipid synthesis and the levels LCB and ceramide species were higher compared to WT cells (Figure 7F). This result is in line with earlier observations showing that the deletion of *ORM1* and *ORM2* strongly de-repressed sphingolipid synthesis (Breslow et al 2010 Nature, Fig. 1b).

Since the enzymatic activity of Lcb1/2 is essential for viability, the reduced levels of LCB and ceramide species are probably just sufficient for the survival of these mutants; an even stronger reduction might be lethal. Conversely, high levels of LCBs and ceramides are toxic. Hence cells tightly control the levels of first metabolic intermediates in SL biosynthesis in a very narrow range. Interestingly, not all LCB or ceramide species were equally affected by Orm2 accumulation. We speculate that this might have to do with differences in the activation of compensatory pathways such as stimulation of ceramide synthase (Lag1/Lac1) by TORC2-Ypk1 signaling (Muir A et al. eLife 2014, PMID: 25279700). Dissecting these specific effects will be interesting, but currently goes beyond the scope of our manuscript.

Collectively, our new results imply that Orm2 degradation restricts the retrieval of Orm2 back the ER and prevents re-association with SPT and repression of sphingolipid biosynthesis. Failure to degrade Orm2 (either in Dsc mutants or Orm2-K25,33R) can cause Orm2 retrieval back to the ER and hence accumulation of Orm2 in the ER and in post-ER compartments. The accumulation of Orm2 increased the interaction with the ER-resident SPT, which ultimately resulted in repression of sphingolipid synthesis. Conversely, constitutive Orm2 export (Orm2-3D) from the ER, resulted in disproportionate de-repression of SPT activity and increased sphingolipid synthesis. Thus sphingolipid homeostasis requires a balanced mechanism that includes Orm2 ER export and its subsequent degradation by endosome and Golgi associated degradation.

Comment: Still related with the sphingolipid levels in the Tul1 mutant, is there any information on how these are affected by the Orm2-3A allele, which appears to localize

predominantly to the ER and less sensitive to Tul1 degradation? Similarly, does this allele have a synthetic growth defect with *Vps4Δ*?

Response: We have addressed the effect of Orm2-3A on LCB and ceramide synthesis above.

The expression of Orm2-3A in *vps4Δ* had a moderate effect on growth (please see Figure A below). We have performed this experiment for the convenience of the reviewer and we would prefer not to include the data in the manuscript. In our view, the conclusions appear to be redundant with Figure 3C where we show that deletion of *ORM2* hardly restored the growth of *tul1Δ vps4Δ* double mutants (Figure 3C).

Figure A. - Figures for Referees not shown.

Comment: The effect of Orm2Δ on Tul1ΔVps4Δ growth defect (Fig 3C) is negligible, suggesting that other Tul1 substrates are more relevant to the growth phenotype. In face of the data presented, I would recommend to tone down the conclusions on the effect of Orm2 on the growth defect Tul1ΔVps4Δ.

Response: we have now changed the tone of the paragraph:

The deletion of *ORM2* hardly restored the growth of *tul1Δ vps4Δ* double mutants (Figure 3C). The *vps4Δ tul1Δ orm2Δ* triple mutants still grew poorly when compared to *vps4Δ orm2Δ* and *tul1Δ orm2Δ* double mutants (Figure 3C). It seemed that, in addition to Orm2 other critical substrates of the Dsc complex were degraded by ESCRT-independent pathways.

Minor points:

Minor Comment: The term pERAD is confusing, as it suggests degradation occurring at the ER. Another nomenclature would be advised.

Response: We agree with the reviewer and have tried to come up with a better name: We now suggest: endosome and Golgi associated degradation: EGAD

Minor Comment: Line 295-297, Gld1 and Vld1 trafficking pathways are vaguely introduced. One-line explaining would be helpful for non-experts.

Response: we have modified the text of this paragraph (p.9. lanes 288 – 298)

Minor Comment: Figure 3C, in *Tul1ΔVps4Δ* double mutants, GFP-Orm2 accumulated in class E compartments, but why is Orm2 depleted from the ER in these mutants? Do the levels of Orm2 differ from *Vps4Δ* cells and to *Tul1ΔVps4Δ*? This should be clarified.

Response: We thank the reviewer for pointing this out. The images presented in the original version of the manuscript focused on showing GFP-Orm2 in class E compartments. They were not ideally suited to evaluate the ER localization. Moreover, the strong GFP-Orm2 signal at the class E compartment made it difficult to visualize the weaker signal at the ER at the same time. We now show images of cells in which the localization of GFP-Orm2 to class E compartments and to the ER can be detected, which is the case in the majority of all cells (Figure 3C). Moreover, based on the reviewer's request, we compared the protein levels of Orm2 in *tul1Δ vps4Δ* double mutants to *vps4Δ*. The protein levels of Orm2 were elevated in *tul1Δ vps4Δ* double mutants compared to *vps4Δ* (Figure EV3B).

Minor Comment: Figure 6E, the levels of Orm2-3A protein appear slightly higher than Orm2 (quantification would be appropriate), as it would be expected from the model proposed. However, the levels still respond to the presence of Tul1. How can this be explained? It would be important to point this out in the text and speculate about possible reasons.

Response: We thank the reviewer for pointing this out. We have indeed missed this result. We have now quantified Orm2 protein levels: Compared to the protein levels of Orm2 in WT cells, the protein levels of Orm2-3A and Orm2-K25,33R and of Orm2 in *tul1Δ* increased by a factor of app. 2 - 3 fold (see Figure EV5D). As the reviewer pointed out, the protein levels of the Orm2-3A mutant increased in *tul1Δ* mutants (app. 1.5 fold). This is not due to an increase in Orm2-3A mRNA levels in the *tul1Δ* mutants (data not shown). How do we interpret these results? Our results show that TORC2-Ypk1 signaling and Orm2 phosphorylation are important for efficient Orm2 ER export and thus initiate degradation (Figure 7). However, we cannot exclude that a small fraction of Orm2-3A can escape the ER and is subsequently degraded by the Dsc complex. At the moment this seems to be the most likely conclusion.

We have added the following paragraph to the result section:

Due to the impaired degradation of Orm2-3A, its protein levels were always higher compared to WT cells (Figure 6E, Figure EV5D). Yet, the protein levels of Orm2-3A still increased moderately in *tul1Δ* mutants (Figure 6E, Figure EV5D). Thus, it seemed possible that a small fraction of Orm2-3A escaped from the ER to Golgi and endosomes where it was degraded by EGAD.

Minor Comment: The references throughout the manuscript are erratic, sometimes absent (e.g. line 84), other times overly used (e.g. line 191-194) and in other instances misused (e.g. line 335 Baldrige & Rapoport 2016- Bodnar & Rapoport 2017 would be more appropriate).

Response: Thank you for pointing out these mistakes. We have fixed it.

Minor Comment: Proteostasis is sometimes broadly used, e.g. line 154.

Response: Thank you for pointing this out. We have reduced the use of ‘proteostasis’. E.g. we have rephrased it to: ‘...collectively maintain eukaryotic membrane protein homeostasis.’ (p. 5, line 149).

Minor Comment: The mass spectrometry results only show data on ceramides. It would be interesting if it also included long chain bases and other sphingolipids.

Response: The analysis of LCBs was very difficult. LCB levels are very low since they are quickly metabolized and their accumulation would be toxic for cells. Nevertheless we managed to measure LCB levels (C18-PHS and C18-DHS). The results are presented in Figure 7F. We could not quantify complex sphingolipids due to the lack of appropriate standards, but we assess complex SL levels in Dsc mutants by established methods for metabolic labeling and thin layer chromatography (Figure 6B).

Minor Comment: Differences were found in Orm2 phosphorylation mediated by Ypk1 versus by Npr1 (Figure 7A and B, line 420-424). These results are not interpreted or discussed.

Response: the following sentence has been added:

In contrast, Npr1 phosphorylation of Orm2, which stimulates complex sphingolipid synthesis (Shimobayashi, Oppliger et al., 2013), was dispensable for Orm2 degradation. The respective Orm2-7A (S9,15,22,29,31A, T18,36A) was still degraded (Figures 7A and 7B).

Minor Comment: The discussion overly focuses on homology analysis of Dsc complex proteins. Perhaps this could be substantially reduced and include the above mentioned points.

Response: we have focused the discussion and reduced the speculations about possible orthologues of the Dsc complex.

Referee #3:

Schmidt et al. present data for a membrane protein degradation pathway that operates primarily in an organelle(s) after the ER (post-ERAD). The pathway was found through a genetic screen for negative synthetic interactions with *vps4*, a mutant in the ESCRT pathway. A key hit in the screen was the Tul1 RING ubiquitin ligase and an associated subunit from the DSC ubiquitination complex. This had previously been shown to ubiquitinate substrates in the Golgi and endosome and cause their degradation via the ESCRT pathway. However, Schmidt et al. find that the model transmembrane substrate, Orm2, that they identified through proteomic analysis was extracted from membranes by Cdc48 and degraded by the 26S proteasome. Disruption of Orm2 ubiquitination and degradation interfered with sphingolipid homeostasis.

This is a very thorough analysis, and although hints of this post-ERAD pathway existed before (Hwang et al. EMBO J. 2016 found that fission yeast SREBP can be degraded through DSC and the proteasome when a cofactor is deleted), the current study makes a much fuller case. Overall, I think this study is suitable for publication in the EMBO Journal with just some modest changes.

Response: We would like to thank the reviewer for the positive comments and the insightful suggestions.

Comment: Localization data are not quantified. The authors should at least state explicitly that the localization phenotypes they see are fully penetrant.

Response: The phenotypes are highly penetrant and always show up in > 70% of all GFP positive cells. The results of the quantifications have been added to the text (p.9. lines 281 – 294 and p. 14, line 447, 461).

Comment: Also, it is known that DSC ligase inhibition in *S. pombe* blocks ER-to-Golgi transport of the DSC complex (see Hwang et al). I was curious if the same occurs in *S. cerevisiae* (not essential to test here but might get at the question of why ERAD E3s apparently cannot compensate for DSC loss).

Response: In budding yeast, the export of Ubx3-mNeonGreen from the ER appeared to be independent of Tull1 (Figure B). Interestingly, it seemed that Ubx3-mNeon was no longer transported into the lumen of the vacuole in *tull1*Δ mutants. We currently try to understand the significance of this observation and provide the results of this experiment for the reviewer here.

Figure B. - Figures for Referees not shown.

Comment: In Fig. 6A, are these differences all statistically significant? I don't see any statistical analysis here, and some of the differences are modest.

Response: Some of the difference are modest and in some cases the p-values are rather high. Yet, we have repeated the experiments 2x with similar trends and extended it to LCB analysis as well (Figure 6A, Figure 7F). To compare the two independent lipid analysis experiments presented in Figure 6A and Figure 7F, we normalized the values for ceramides to Cer44:0;4 and for LCBs to C18-PHS in WT cells. This was necessary due to two different lipid analysis protocols (one of which better suited to also extract LCBs), new pre-separation columns/gradients and difference in the batches for the ceramide standards and their detection. Regardless of the different absolute values, the relative differences between WT cells and the mutants were comparable (Figure 6A and Figure 7F). In each experiment, *tull1*Δ mutants and cells expressing Orm2-3A or Orm2-K25,33R had lower levels of LCB and ceramide species than WT cells (Figure 6A and Figure 7F). These results are comparable to the defects in LCB and ceramide synthesis caused by the strong over-expression of Orm2 or by an Orm1/2 double phospho-mutant strain (Breslow et al., 2010, Nature, Fig. 1b and Fig. 5d in their paper). In contrast, Orm2-3D de-repressed sphingolipid synthesis and the levels LCB and ceramide species were higher compared to WT cells (Figure 7F). This result is in line with earlier observations showing that the deletion of *ORM1* and *ORM2* strongly de-repressed sphingolipid synthesis (Breslow et al 2010 Nature, Fig. 1b).

Since the enzymatic activity of Lcb1/2 is essential for viability, the reduced levels of LCB and ceramide species are probably just sufficient for the survival of these mutants; an even stronger reduction might be lethal. Conversely, high levels of LCBs and ceramides are toxic. Hence cells tightly control the levels of first metabolic intermediates in SL biosynthesis in a very narrow range. Interestingly, not all LCB or ceramide species were equally affected by Orm2 accumulation. We speculate that this might have to do with differences in the activation

of compensatory pathways such as stimulation of ceramide synthase (Lag1/Lac1) by TORC2-Ypk1 signaling (Muir A et al. eLife 2014, PMID: 25279700). Dissecting these specific effects will be interesting, but currently goes beyond the scope of our manuscript.

Comment: Line 404, Figure 6C-D: Is the fraction of Orm2 in the phosphorylated state increased relative to unmodified protein?

Response: Based on WB quantification, the levels of phosphorylated Orm2 were always increased in *tull1Δ* mutants or in cells expressing Orm2-K25,33R relative to WT cells (approximately 3-fold, Figure EV5A). This is consistent with TORC2-Ypk1 activation caused by lower SL lipid levels. Of course the total protein levels of Orm2 are also increased in these mutants (Figure EV5D). Overall there is more phosphorylated Orm2 in these mutants, but still they fail to de-repress SL synthesis. Due to known technical limitations of Phostag gels (unphosphorylated proteins are more efficiently transferred during the Western blot) we cannot provide reliable WB quantification of the fraction of phosphorylated Orm2 relative to non-phosphorylated Orm2.

Comment: To more convincingly rule out a role for known ERAD pathways in Orm2 degradation, it would be worth testing a *hrd1 doa10* double (or possibly *hrd1 doa10 asi1* triple) mutant.

Response: Orm2 is still degraded in *doa10Δ hrd1Δ* double mutants and in *doa10Δ hrd1Δ asi1Δ* triple mutants (Figure 4A,B and Figure EV4B).

Comment: The use of commas between allele names when referring to multiple mutant yeast strains (not what most yeasties would recommend) makes the text hard to follow in places, such as lines 308-9.

Response: We have fixed this, thank you

Comment: The references are in general very thorough, but for one ERAD E3 ligase, MARCH6/TEB4 (Doa10 in yeast), there are no/almost no references.

Response: Thank you for pointing this out. We have included this now in the introduction.

Reviewer 2 also recommended to avoid the name pERAD, as it may suggest ER based processes. We agree that the name could be confusing and therefore suggest another (hopefully better) name: endosome and Golgi associated degradation – EGAD.

Thank you for submitting your revised manuscript for our consideration. It has now been seen once more by two of the original reviewers (see comments copied), and I am happy to let you know that both are satisfied with the revisions and have no more objections towards publication in The EMBO Journal.

After some final editorial modifications, we shall therefore be ready to proceed with formal acceptance and publication of the study!

REFEREE REPORTS

Referee #2:

The authors did a great job during revision and all my concerns have been addressed. I am strongly in favor of the publication of this manuscript.

Referee #3:

The responses to my relatively minor suggestions for improvement, mostly issues about quantitation and reproducibility, are satisfactory. And I like "EGAD" - it may help to revive an obsolete but charming English expletive.

Corresponding Author Name: David Teis
Journal Submitted to: EMBO Journal
Manuscript Number: EMBOJ-2018-101433